# Atomically dispersed iridium catalysts on silicon photoanode for efficient photoelectrochemical water splitting

Sang Eon Jun [1], Youn-Hye Kim[2], Jaehyun Kim[1], Woo Seok Cheon[1], Sungkyun Choi[1], Jinwook Yang[1], Hoonkee Park[1], Hyungsoo Lee [3], Sun Hwa Park[4], Ki Chang Kwon [4], Jooho Moon [3] ✉, Soo-Hyun Kim [5] ✉ & Ho Won Jang [1,6] ✉

Stabilizing atomically dispersed single atoms (SAs) on silicon photoanodes for photoelectrochemical-oxygen evolution reaction is still challenging due to the scarcity of anchoring sites. Here, we elaborately demonstrate the decoration of iridium SAs on silicon photoanodes and assess the role of SAs on the separation and transfer of photogenerated charge carriers. NiO/Ni thin film, an active and highly stable catalyst, is capable of embedding the iridium SAs in its lattices by locally modifying the electronic structure. The isolated iridium SAs enable the effective photogenerated charge transport by suppressing the charge recombination and lower the thermodynamic energy barrier in the potential-determining step. The Ir SAs/NiO/Ni/ZrO$_2$/n-Si photoanode exhibits a benchmarking photoelectrochemical performance with a high photocurrent density of 27.7 mA cm$^{-2}$ at 1.23 V vs. reversible hydrogen electrode and 130 h stability. This study proposes the rational design of SAs on silicon photoelectrodes and reveals the potential of the iridium SAs to boost photogenerated charge carrier kinetics.

Photoelectrochemical (PEC) tandem devices, highly desirable for converting intermittent solar energy into sustainable hydrogen fuel, are promising architectures toward zero-carbon society[1–3]. Si has been widely investigated for photoelectrode owing to its excellent charge carrier mobility, long carrier diffusion length, earth abundance, and wide-range solar spectrum absorption[4–6]. Nevertheless, the poor catalytic activity, chemical corrosion in aqueous electrolyte, and negative valence band position limit its practical application[7,8]. Hence, it is necessary to apply the photoelectrochemical catalysts having large amounts of active sites with complete coverage of photoelectrodes. Specifically, the atomically modified thin film on Si photoanodes can serve as PEC catalysts boosting an accelerated surface reaction kinetics with the high stability required for efficient light harvesting[9]. In addition, metal-insulator-semiconductor (M-I-S) structures enable the devices to maximize the photovoltage by eliminating Fermi level pinning effect and increasing barrier height of semiconductor[10–12]. Though lots of efforts have been made to apply transition metal and noble metal-based PEC catalysts to Si photoanodes and insert thin insulating layer for high photovoltage, few devices meet the criteria of PEC performance for practical use.

Single-atom catalysts (SACs) have been recently investigated in electrochemical (EC) reaction to not only maximize the atomic

[1]Department of Materials Science and Engineering, Research Institute of Advanced Materials, Seoul National University, Seoul 08826, Republic of Korea. [2]School of Materials Science and Engineering, Yeungnam University, Gyeongsan, Gyeongbuk 38541, Republic of Korea. [3]Department of Materials Science and Engineering, Yonsei University, Seoul 03722, Republic of Korea. [4]Interdisciplinary Materials Measurement Institute, Korea Research Institute of Standards and Science, Daejeon 34113, Republic of Korea. [5]Graduate School of Semiconductor Materials and Devices Engineering, Ulsan National Institute of Science and Technology, 50 UNIST-gil, Ulju-gun, Ulsan 44919, Republic of Korea. [6]Advanced Institute of Convergence Technology, Seoul National University, Suwon 16229, Republic of Korea. ✉e-mail: jmoon@yonsei.ac.kr; soohyunsq@unist.ac.kr; hwjang@snu.ac.kr

efficiency of noble metal catalysts but also introduce unconventional geometric and electronic structures[13–16]. The modification of the chemical state of single atoms induced by electronic interactions with the support matrix plays essential role in improving catalytic activity and stability[15,17,18]. Despite their unique properties and capability to accelerate catalytic activity, few researches on SACs have been reported in PEC application due to the difficulty lying in creating anchor sites on photoelectrodes and analyzing the coordination environment of single atoms. Furthermore, there is no case of observing the photogenerated charge carrier kinetics on SACs applied to the photoelectrodes. Unlike the EC system, it is important to assess the capability of single atoms to reduce surface recombination of photoinduced charges as facile transport to the electrolyte determines the PEC performance of the device[19]. If it can be numerically analyzed, it will be of great help to the subsequent research on atomically dispersed catalysts for PEC application with low cost, high efficiency, and long-term durability.

Here, we report an atomic-scale-architectured Ir SAs/NiO/Ni/ ZrO$_2$/n-Si photoanode device with drastically enhanced catalytic activity and stability. The NiO/Ni, which is highly catalytic and stable in alkaline condition, intrinsically has lots of Ni vacancies suitable for stabilization of single atoms[20–24]. As a result, it is possible to utilize the thin-film NiO/Ni as both a PEC catalyst and anchoring layer to capture metal atoms. A single cycle atomic layer deposition (ALD), relying on molecular-level self-limiting surface reaction, is utilized to simultaneously convert the surface of Ni into NiO and deposit atomically dispersed Ir catalysts[25–27]. The existence of Ir single atoms is identified by the aberration-corrected scanning transmission electron microscopy (STEM) and their unique electronic environment is analyzed by X-ray photoelectron spectroscopy (XPS) and X-ray absorption spectroscopy (XAS). Furthermore, the ZrO$_2$ insulating layer with 1 nm thickness enables to construct M-I-S structure and alleviates the Fermi level pinning effect, leading to the enlarged photovoltage. The Ir SAs/ NiO/Ni/ZrO$_2$/n-Si exhibit a high-record PEC performance with the photocurrent density of 27.7 mA cm$^{-2}$ at 1.23 V versus a reversible hydrogen electrode (RHE) and 130 h stability. The intensity-modulated photocurrent spectroscopy (IMPS) and electrochemical impedance spectroscopy (EIS) reveal that the Ir single atoms effectively suppress the photogenerated charge carrier recombination at the surface and boost the charge transport to the electrolyte. Compared to Ir nanoclusters and film catalysts, Ir single atoms show the highest charge transfer efficiency ($\eta_{trans}$) and the lowest charge transfer resistance ($R_{ct}$). Also, by employing density functional theory (DFT) calculations, it is identified that the isolated Ir atoms act as catalytically active centers lowering the thermodynamic energy barrier of the potential-determining step and allowing the photoinduced holes easily transfer to the adsorbed OH. This study contributes to the elaborate synthesis of a thin film with single-atom catalysts and demonstrates the huge potential of the Ir single atoms to boost photogenerated charge carrier kinetics.

## Results

### Preparation and characterization of Ir single atoms anchored on NiO/Ni/ZrO$_2$/n-Si photoanodes

Before anchoring Ir single atoms, the M-I-S junction is formed by inserting ZrO$_2$ layer between n-Si semiconductor and Ni metal layer. It reduces the interface states at the surface of Si and the Fermi level pinning effect is eliminated, resulting in the enhancement of photovoltage (Supplementary Fig. 1). To construct the NiO/Ni thin film, Ni layer with the thickness of 2 nm was deposited and the surface of Ni is oxidized by oxygen supplied as a secondary precursor vapor for Ir deposition during the ALD process. Then, atomically dispersed Ir single atoms are anchored into the NiO lattice via a single cycle ALD process, utilizing tricarbonyl (1,2,3-η)-1,2,3-tri(tert-butyl)-cyclopropenyl iridium (C$_{18}$H$_{27}$IrO$_3$ or TICP) and O$_2$ as a precursor vapor[28]. The cross-sectional high-resolution TEM image in Fig. 1a shows ZrO$_2$ and

NiO/Ni thin film layers deposited on n-type Si photoanode. The amorphous ZrO$_2$ with 1 nm thickness can act as both hole transporting layer and corrosion-resistant layer. The atomic and lattice structure of NiO were identified by HR-TEM images and the fast Fourier transformation (FFT) pattern in Fig. 1b. The lattice fringes with an interplanar distance of 0.199 nm and 0.242 nm correspond to (200) and (111) crystallographic plane[20,29]. As shown in the inset of Fig. 1b, the FFT pattern presents the diffraction spots for (200) and (111). Atomically dispersed Ir single atoms embedded on NiO thin film are clearly observed and discerned by aberration-corrected high-angle annular dark field scanning TEM (HAADF-STEM) in Fig. 1c and Supplementary Fig. 2.

It is revealed that homogeneous Ir atoms exist in isolation without forming aggregated nanoclusters and nanoparticles. In Fig. 1d, e, the atomic line profiles were demonstrated and Ir atoms are distinguished by the brighter spots with high signal intensity. It was clearly confirmed that the Ir atoms were located at exactly the same positions of Ni atoms with high periodicity, exhibiting that the original Ni sites were substituted by Ir atoms. The constituent elements of Ir SAs/NiO/Ni catalysts are analyzed by EDS mapping images in Fig. 1f. The results further prove that Ir atoms are uniformly dispersed on NiO/Ni thin film. In the case of 1 cycle ALD, Ir single atoms were formed, whereas in the case of multiple cycles (25, 100, 200 cycles), nanoclusters (NCs), film, and much thicker film (Ir film-T) were synthesized (Supplementary Fig. 3). After the saturation of Ni vacancies, the aggregation is promoted due to the increased surface free energy derived from the unsaturated coordination of Ir[30]. The X-ray diffraction (XRD) patterns of the samples are provided in Supplementary Fig. 4. Only for Ir film and Ir film-T, the characteristic peaks of Ir are observed at 41° and 48°, which correspond to the (111) and (200) facets of Ir, respectively. From this result, we can see that Ir SAs and NCs exist without crystallinity. To clarify the amounts of photons absorbed and reflected by the catalyst layers, the optical transmittance spectra are measured in the wavelength range from 300 to 1200 nm in Supplementary Fig. 5. The ZrO$_2$ is wide-bandgap material, so all the light passes through it. For the Ni layer, about 20% of the light is mostly reflected. After a large amount of Ir deposition (Ir film and Ir film-T), the transmission of light is remarkably suppressed due to the light reflection, leading to low photon absorption of Si photoanodes. On the contrary, the light transmittance is increased after the deposition of Ir SAs and NCs. It is attributed to the conversion of a portion of Ni into NiO semiconductor and negligible light reflection by Ir. To explore the chemical composition and electronic states of Ir SAs, NCs, and film deposited on the NiO layer, X-ray photoelectron spectroscopy analysis is carried out and wide scans are presented in Supplementary Fig. 6. Figure 2a shows the deconvolution of the Ir 4$f$ core level spectra of Ir SAs, NCs, and film. For better visualization and clarification of XPS results, the magnified and simplified version of Fig. 2a is provided in Supplementary Fig. 7. For both Ir NCs and film, the peaks are deconvoluted into three spin-orbit splitting doublets at the same binding energy. One pair of doublets at 64.8 eV/61.7 eV (Ir$^{4+}$4$f_{5/2}$ and Ir$^{4+}$4$f_{7/2}$) and another pair of doublets at 64.4 eV/61.4 eV (Ir$^{3+}$4$f_{5/2}$ and Ir$^{3+}$4$f_{7/2}$) are ascribed to iridium oxide[13,31]. The other pair of doublets at 63.8 eV/60.8 eV (Ir$^0$4$f_{5/2}$ and Ir$^0$4$f_{7/2}$) corresponds to metal Ir species. Meanwhile, Ir SAs exhibit only one doublet at 64.6 eV and 61.6 eV near the energy of iridium oxide. It indicates that atomically dispersed Ir atoms on NiO lattice exist mainly at the +3-+4 valence states apart from metallic Ir. The modification of the chemical state of Ir is attributed to the charge transfer between NiO and Ir species. Furthermore, the Ir-O peak in O 1$s$ spectra of Ir SAs implies the existence of the interaction between Ir and NiO (Supplementary Fig. 8).

To further probe the electronic states and coordination environment of the Ir SAs/NiO catalyst, X-ray absorption near-edge structure (XANES) spectroscopy and extended X-ray absorption fine structure (EXAFS) spectroscopy are carried out. Commercial IrO$_2$ powder, Ir

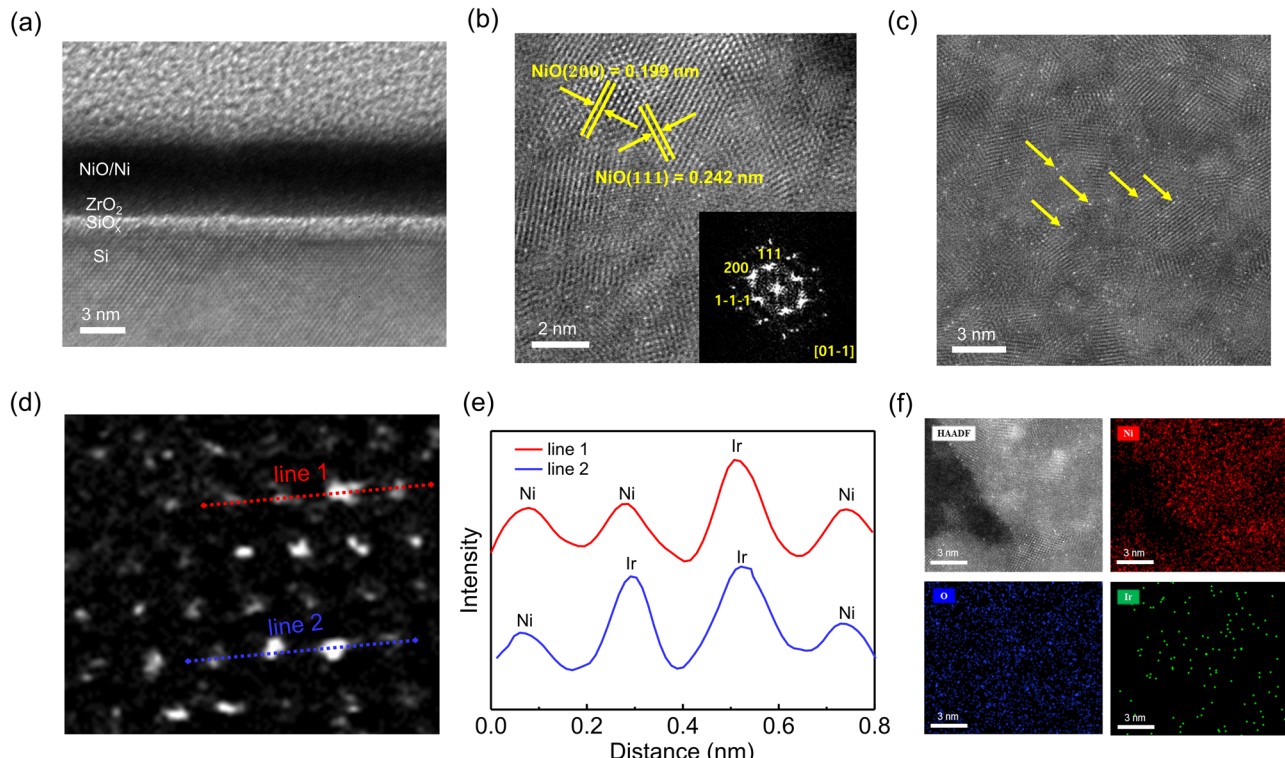

**Fig. 1 | Transmission electron microscope analysis of Ir SAs/NiO/Ni/ZrO₂/n-Si. a** Cross-sectional TEM image of Ir SAs/NiO/Ni/ZrO₂/n-Si photoanodes. **b** High-resolution top-view TEM image of Ir SAs/NiO surface and corresponding fast Fourier transformation (FFT) pattern. **c** Top-view HAADF-STEM image of Ir SAs/NiO thin-film catalyst. **d, e** Intensity profiles (line 1 and 2) showing the intensity difference between Ni and Ir atoms. **f** STEM-EDS elemental mapping of Ir SAs/NiO thin-film.

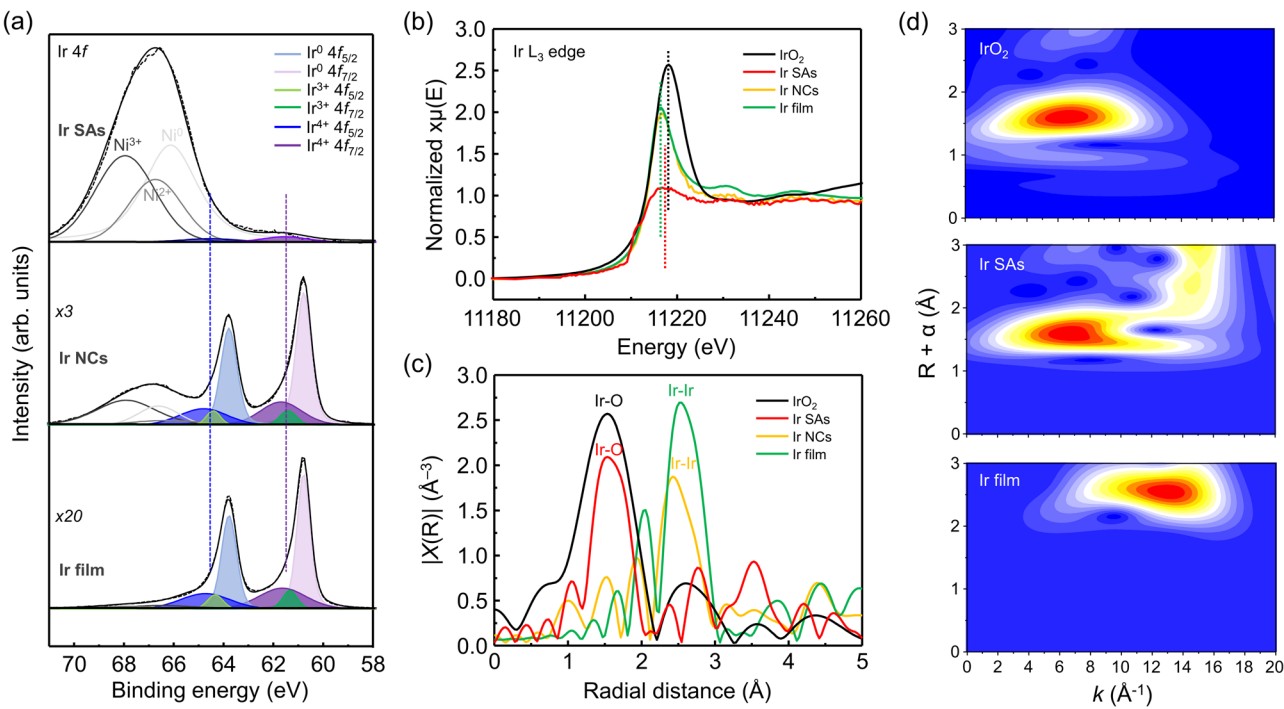

**Fig. 2 | X-ray photoelectron spectroscopy and X-ray absorption analysis for atomic structure characterizations. a** Ir 4*f* spectra of NiO/Ni/ZrO₂/n-Si photoanodes deposited with Ir SAs, NCs, and film. **b** Ir L₃-edge XANES spectra of IrO₂, Ir SAs, NCs, and film. **c** Fourier transform EXAFS spectra of IrO₂, Ir SAs, NCs, and film. **d** Wavelet transforms (WT) for the k³-weighted EXAFS signals of IrO₂, Ir SAs, and Ir film.

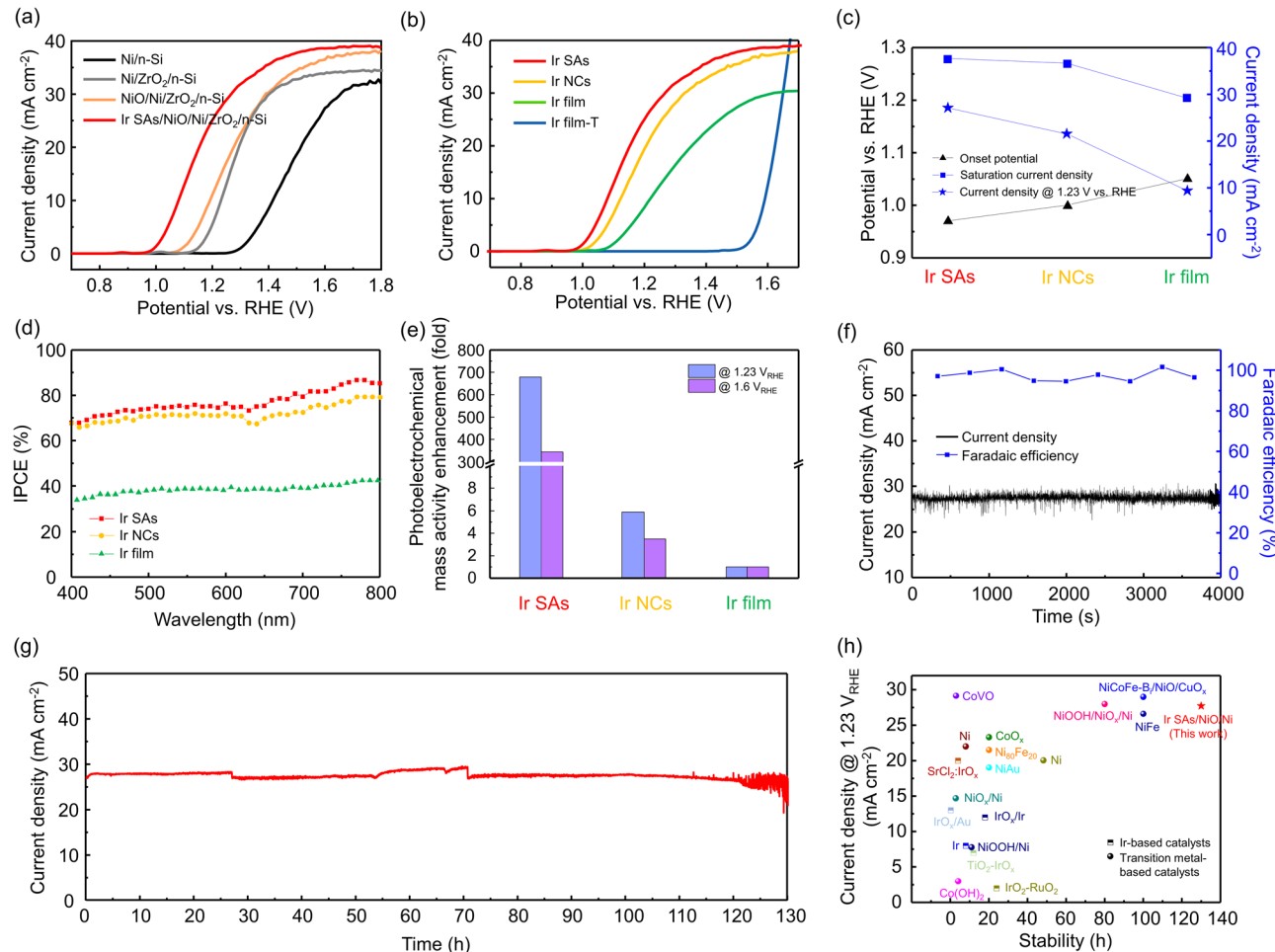

**Fig. 3 | Photoelectrochemical performance of Ir SAs/NiO/Ni/ZrO₂/n-Si. a** Linear sweep voltammograms (LSVs) of the Ni/n-Si, Ni/ZrO₂/n-Si, NiO/Ni/ZrO₂/n-Si, and Ir SAs/NiO/Ni/ZrO₂/n-Si photoanodes measured in 1 M NaOH electrolyte. **b** LSVs of NiO/Ni/ZrO₂/n-Si photoanodes deposited with Ir SAs, NCs, film, and much thicker film. **c** Comparison of the onset potential, saturation current density, current density at 1.23 V vs. RHE of the photoanodes. **d** Incident-photon-to-current conversion efficiency (IPCE) measurements. **e** Comparison of photoelectrochemical mass activities among catalysts at 1.23 and 1.6 V_RHE. **f** Faradaic efficiency of Ir SAs/NiO/Ni/ZrO₂/n-Si photoanode during chronoamperometry measurements at 1.23 V_RHE. **g** Long-term stability test during 130 h. **h** Comparison of Ir SAs/NiO/Ni catalysts with recently reported Ir-based and transition metal-based PEC catalysts.

NCs, and Ir film were compared as benchmarks. For the Ir L₃-edge XANES spectra in Fig. 2b, the white line peak position of Ir SAs is located closer to that of IrO₂ than that of metallic Ir, implying the oxidation state of Ir in Ir SAs/NiO is near 4+, which agrees with the result of XPS. The Fourier transformed (FT)-EXAFS spectrum of Ir SAs in Fig. 2c shows a prominent peak at the radial distance of 1.54 Å. It is consistent with a peak of IrO₂ derived from the Ir-O scattering path. However, the spectra of Ir NCs and film exhibit the peaks near 2.5 Å, corresponding to Ir−Ir interactions. No typical peak representing Ir−Ir scattering is observed for Ir SAs, indicating Ir atoms are dispersed in isolation. In Fig. 2d, the wavelet transforms (WT) for the EXAFS signals, which enable to precisely discriminate backscattering atoms with high resolution in not only $R$ space but also energy space, are demonstrated to further explore the atomic dispersion of Ir SAs[32]. For both IrO₂ and Ir SAs, only one intensity maximum was observed at 6.6 Å⁻¹ derived from Ir-O scattering, while Ir film shows an intensity maximum at 12.9 Å⁻¹. From these results, it is concluded that Ir single atoms deposited by a single cycle ALD totally interact with NiO support with the complete absence of Ir−Ir bonding.

## Photoelectrochemical OER performance

The photoelectrochemical OER activities of the fabricated photoanodes are measured under a simulated air mass 1.5 G solar illumination

using a standard three-electrode system with 1 M NaOH electrolyte. Linear sweep voltammograms (LSVs) of Ni/n-Si, Ni/ZrO₂/n-Si, NiO/Ni/ZrO₂/n-Si, and Ir SAs/NiO/Ni/ZrO₂/n-Si photoanodes are shown in Fig. 3a. When only Ni film is deposited on n-Si photoanode, the high value of onset potential is shown due to low photovoltage derived from the Fermi level pinning effect. To achieve high photovoltage, the interfacial energetics are manipulated by applying the ZrO₂ tunneling oxide layer, leading to the formation of metal-insulator-semiconductor junctions. As a result, the onset potential shifts toward the cathodic direction with the value of 1.14 V versus RHE to reach 1 mA cm⁻² and the saturation photocurrent density was increased by 2 mA cm⁻². To further support it, Mott-Schottky measurements are conducted to acquire the flat band potential (E_fb) of Ni/n-Si and Ni/ZrO₂/n-Si photoanodes in Supplementary Fig. 9. The negative shift in flat band potential of Ni/ZrO₂/n-Si (−0.7 V vs. Ag/AgCl) relative to that of Ni/n-Si (−0.4 V vs. Ag/AgCl) proves the enlarged photovoltage resulted from the elimination of Fermi level pinning effect. As the thickness of this insulating layer becomes thicker, the movement of the holes is restricted, resulting in the deteriorated PEC performance (Supplementary Fig. 10). The thermally oxidized Ni surface in NiO/Ni/ZrO₂/n-Si induces enhanced catalytic activity and photon absorption, which is confirmed by lowered onset potential and increased saturation current. The Ir SAs/NiO/Ni/ZrO₂/n-Si photoanode exhibits dramatically

enhanced photoelectrochemical performance with the onset potential of 0.97 V vs. RHE and the current density of 27.7 mA cm$^{-2}$ at 1.23 V vs. RHE. It is attributed to the capability of Ir SAs that enable the facile charge transport by suppressing the charge recombination and boost the catalytic surface reaction. To verify the role of each layers, the LSV curves of NiO/Ni/ZrO$_2$/n-Si, Ir ALD-1 cyc./ZrO$_2$/n-Si, and Ir SAs/NiO/Ni/n-Si are provided in Supplementary Fig. 11. For the photoelectrode without NiO/Ni, Ir atoms are not anchored on the surface, resulting in no photoelectrochemical catalytic activity. Also, the role of ZrO$_2$ increasing the photovoltage can be seen through the photoanode where ZrO$_2$ does not exist. We also fabricated NiFe-based photoanodes (NiFe/n-Si and Ir (ALD-1cyc.)/NiFe/n-Si) as it is well known that NiFe alloy shows a higher catalytic activity than Ni. However, their PEC performance did not last long due to the leaching of Fe[33], as shown in Supplementary Fig. 12.

In Fig. 3b and Supplementary Fig. 13, the photoanode with Ir SAs shows the lowest onset potential and the highest photocurrent density over the whole potential range among all samples, implying that Ir SAs exhibit the highest catalytic activity and photon harvesting. When the Ir film becomes too thick (Ir film-T), the photogenerated electrons can no longer participate in the reaction and only electrochemical water oxidation occurs. The specific values of onset potential, saturation current density, and current density at 1.23 V versus RHE of all samples are provided in Supplementary Fig. 14, Fig. 3c, and Supplementary Table 1. In Fig. 3c, the onset potential, determined by photogenerated charge transport and catalytic activity, decreased to 0.97 V versus RHE with the Ir single atoms. It indicates that the photogenerated holes of Ir SAs/NiO/Ni/ZrO$_2$/n-Si can easily reach the electrolyte interface and participate in OER reaction actively compared to Ir NCs and film. Also, the saturation current density is considerably increased to 38 mA cm$^{-2}$, implying that the light reflection induced by the metallic nature of Ir is suppressed due to atomically dispersed morphology. As a result, the photocurrent density of 27.7 mA cm$^{-2}$ is achieved at 1.23 V versus RHE by applying Ir single-atom catalysts. For the electrochemical (EC) measurements in Supplementary Fig. 15, the same tendency is shown in catalytic activity. To precisely determine the photovoltage, Ir SAs/NiO/Ni/ZrO$_2$ is deposited on highly doped p$^{++}$-Si and the LSV curve is obtained in Supplementary Fig. 16. When comparing the onset potential of Ir SAs/NiO/Ni/ZrO$_2$/n-Si and Ir SAs/NiO/Ni/ZrO$_2$/p$^{++}$-Si, it is confirmed that the photovoltage of the photoanode is 550 mV.

The performance of photoanodes to convert the incident light to electrical current density is analyzed by the incident photon-to-current conversion efficiency (IPCE) in Fig. 3d. It is measured from 400 to 800 nm of wavelength and 1.23 V versus RHE is applied. For the photoanode with Ir film, it shows the efficiency of 40% on average over the entire visible light wavelength. The efficiency is significantly increased by introducing Ir nanoclusters, and it reaches up to 75% on average when the single atom Ir catalysts are anchored on the photoanode. The efficiencies of about 40, 65, and 0% are measured for NiO/Ni/ZrO$_2$/n-Si, Ir SAs/NiO/Ni/n-Si, and Ir ALD-1 cyc./ZrO$_2$/n-Si, respectively, in Supplementary Fig. 17. The mass activity is a crucial parameter to evaluate the intrinsic catalytic activity of single atom catalysts quantitatively. For the first time to our knowledge, photoelectrochemical mass activity is analyzed to determine the contribution of Ir atoms to photocurrent per mass depending on the morphology of Ir catalysts. The photocurrent density at a given potential is divided by the mass of Ir measured by inductively coupled plasma-mass spectrometry (ICP-MS) as provided in Supplementary Table 2. Consequently, the PEC mass activity of Ir SAs/NiO/Ni/ZrO$_2$/n-Si, at the potentials of 1.23 and 1.6 V versus RHE, are 115 and 98.5 times higher than that of Ir NCs/NiO/Ni/ZrO$_2$/n-Si, and 679 and 344.6 times higher than that of Ir film/NiO/Ni/ZrO$_2$/n-Si. Furthermore, the Faradaic efficiency of Ir SAs/NiO/Ni/ZrO$_2$/n-Si photoanode is obtained by a gas chromatography measurement in Fig. 3f. During chronoamperometry measurement, the evolved oxygen

gas is collected and almost 100% Faradaic yield is acquired. In Fig. 3g, the long-term stability of as-fabricated Ir SAs/NiO/Ni/ZrO$_2$/n-Si photoanode is examined by chronoamperometry at applied voltage of 1.23 V versus RHE. It demonstrates the remarkably stable PEC performance with 130 h, which is an encouraging result considering that Si photoelectrode is highly vulnerable to an alkaline environment. It is attributed to a chemically robust NiO/Ni catalyst and its capability to stabilize and activate Ir single atoms through strong interactions. Before the degradation of performance, the chemical state of Ir single atoms remained unchanged (Supplementary Fig. 18). In Fig. 3h, Supplementary Tables 3, 4, the comparison of Ir SAs/NiO catalyst with recently reported Ir-based and transition metal-based PEC catalysts is summarized[5,10,12,20,34–50]. They imply that Ir SAs/NiO/Ni is one of the best photoelectrochemical catalysts showing the highest PEC catalytic activity and stability. Even though an extremely small amount of Ir is used in this work, the PEC performance is much better than that of other catalysts with high loading of Ir. To analyze the pH dependence of as-fabricated samples, the LSV curves of Ir SAs, NCs, and film deposited on NiO/Ni/ZrO$_2$/n-Si photoanodes are obtained in acidic condition (0.5 M H$_2$SO$_4$) (Supplementary Fig. 19). Since the Ni-based thin film is highly vulnerable to this environment, Ir SAs/NiO/Ni/ZrO$_2$/n-Si and Ir NCs/NiO/Ni/ZrO$_2$/n-Si photoanodes lost catalytic effects and can not serve as photo-harvesting devices. Meanwhile, Ir film is stable in acidic condition and protects the NiO/Ni from being etched, showing a substantial photoelectrochemical catalytic activity.

## Frequency-domain analysis for photogenerated charge carrier kinetics

In optoelectronics, both light intensity and applied current can be modulated through a sinusoidal perturbation on different time scales. Intensity-modulated photocurrent spectroscopy (IMPS) measures the periodic modulation of the photocurrent in response to a small sinusoidal perturbation of the light intensity, while electrical impedance spectroscopy (EIS) estimates an electrical impedance in relation to a perturbation of an alternating current[51]. These frequency-modulated spectroscopies are powerful tools to reveal the photogenerated charge carrier kinetics by identifying the constants of charge transfer, recombination, and interfacial resistances[52]. In the IMPS technique, a modulated response of the photocurrent with a phase shift depending on perturbed light intensity can be defined as the frequency-dependent photocurrent admittance, $Y(\omega)$ expressed as:

$$Y(\omega) = \frac{J(\omega)}{L(\omega)} = \frac{J_0 \sin(\omega t + \phi)}{L_0 + L' \sin(\omega t)} = Y_0 \frac{\sin(\omega t + \phi)}{\sin(\omega t)} = Y_0 (\cos\phi + j\sin\phi) \quad (1)$$

where $J(\omega)$ is a modulated response signal generated by the modulated light intensity $L(\omega)$[51,53]. Consequently, it is represented by the combination of the real and imaginary parts. By plotting the imaginary part versus the real part, the Nyquist plot can be obtained. To interpret this plot, a model based on classical semiconductor electrochemistry is introduced. The equation is given as:

$$j(\omega) = I_0 \frac{k_{trans} + i\omega \left[ C_{sc} C_H / C_{sc} (C_{sc} + C_H) \right]}{k_{trans} + k_{rec} + i\omega} \left( \frac{1}{1 + i\omega\tau} \right) \quad (2)$$

where $I_0$ is the amplitudes of the photogenerated hole current. The $k_{trans}$ is the charge transfer constant, $k_{rec}$ is the charge recombination constant, $C_{sc}$ and $C_H$ are the capacitances of the space charge region and Helmholtz layer, respectively. $\tau$ is the time constant. Assuming that $\tau$ (= RC) is much smaller than the $k_{trans}$ and $k_{rec}$, the above equation can be written as:

$$j(\omega) = I_0 \left( \frac{k_{trans} + i\omega}{k_{trans} + k_{rec} + i\omega} \right) \quad (3)$$

Using both this equation and obtained Nyquist plots with low-frequency limit ($\omega \to 0$), the charge transfer efficiency can be expressed as:

$$\eta = \frac{k_{trans}}{k_{trans} + k_{rec}} = \frac{\text{Low frequency intercept with the real axis (LFI)}}{\text{High frequency intercept with the real axis (HFI)}} \quad (4)$$

In addition, the average transport time ($\tau_t$) for photoinduced charges can be estimated as:

$$\tau_t = \frac{1}{k_{trans} + k_{rec}} = \frac{1}{2\pi f_{max}} \quad (5)$$

where $f_{max}$ is the frequency for which the value of the imaginary part reaches its maximum. Even though these equations are derived from a classical model based on a bare semiconductor being in contact with the electrolyte, they also can be applied into the photoelectrodes having numerous interfaces with the assumption[54] (Supplementary Fig. 20). In Fig. 4a, IMPS Nyquist plots displayed by the complex photocurrent of Ir SAs, Ir NCs, and Ir film photoanodes at the applied voltage of 1.23 V versus RHE are shown. In addition, the plots of the frequency-dependent imaginary photocurrent are provided in

Fig. 4b. To acquire $\eta_{trans}$, $k_{trans}$, and $k_{rec}$, we will extract two pieces of information from Fig. 4a, b. (1) The charge transfer efficiency ($\eta_{trans}$), represented by $k_{trans}/(k_{trans} + k_{rec})$, can be obtained by the ratio of real photocurrents at the low-frequency and high-frequency intercepts. (2) The combined rate of charge transfer and recombination, represented by ($k_{trans} + k_{rec}$), is calculated by $2\pi f_{max}$. With the detailed calculations in the supplementary information, it is possible to obtain $k_{trans}$ and $k_{rec}$ of Ir SAs/NiO/Ni/ZrO$_2$/n-Si (146.2 s$^{-1}$, 97.46 s$^{-1}$), Ir NCs/NiO/Ni/ZrO$_2$/n-Si (123.22 s$^{-1}$, 189.52 s$^{-1}$), and Ir film/NiO/Ni/ZrO$_2$/n-Si (9.75 s$^{-1}$, 233.91 s$^{-1}$) at 1.23 V versus RHE. It is noteworthy that the value of $k_{trans}$ exceeds that of $k_{rec}$ only for the photoanode to which Ir single atoms are applied. These results demonstrate the crucial role of Ir single atoms as photoelectrochemical catalysts, which prevents photogenerated charge carrier recombination at the surface and shows an outstanding OER catalytic activity. The IMPS measurements at 1.0 and 1.4 V versus RHE are also analyzed in Supplementary Fig. 21. In Fig. 4c, d, Supplementary Fig. 22, and Supplementary Table 5, the $\eta_{trans}$, $k_{trans}$, and $k_{rec}$ at different applied potentials are provided. In Fig. 4c, d, Ir SAs/NiO/Ni/ZrO$_2$/n-Si photoanode shows the highest charge transfer efficiency at all bias due to the large value of $k_{trans}$ compared to $k_{rec}$. It is the quantitative evidence for the facile hole transport and suppressed surface recombination around Ir single atoms across the Helmholtz layer.

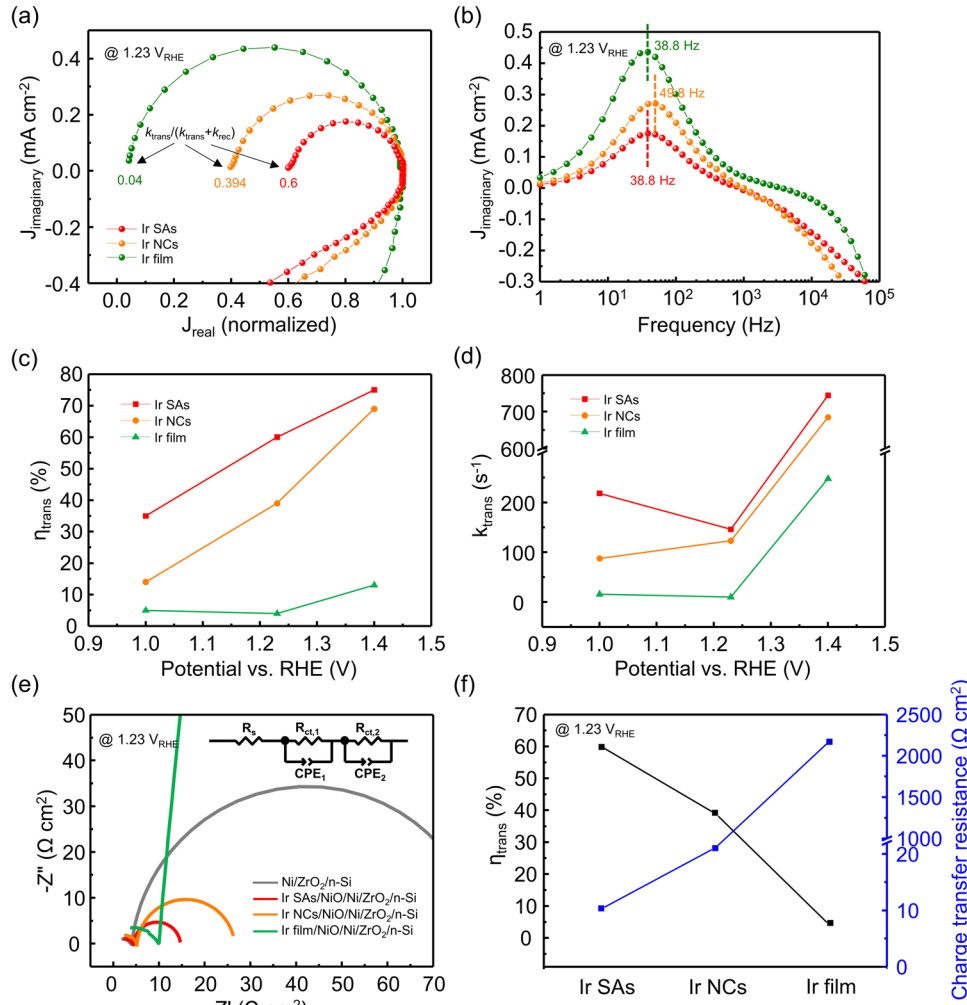

**Fig. 4 | Frequency-domain analysis for photogenerated charge carrier dynamics. a** Intensity-modulated photocurrent spectroscopy (IMPS) Nyquist plots showing the imaginary photocurrent vs. the normalized real photocurrent at 1.23 V$_{RHE}$. **b** Frequency-dependent imaginary photocurrent plots. **c** Charge transfer efficiency and **d** Charge transfer constant vs. potential. **e** Electrochemical impedance spectroscopy (EIS) plots. **f** The plots of charge transfer efficiency from the IMPS data and charge transfer resistance from EIS data at 1.23 V versus RHE.

It is possible to obtain the value of interfacial resistances applied to photogenerated charges at each interface of the photoelectrode by EIS measurement. In Fig. 4e, EIS data of Ni/ZrO₂/n-Si, Ir SAs/NiO/Ni/ZrO₂/n-Si, Ir NCs/NiO/Ni/ZrO₂/n-Si, and Ir film/NiO/Ni/ZrO₂/n-Si are represented by Nyquist plots and fitted to a simplified equivalent circuit which is composed of charge transfer resistance ($R_{ct}$) and constant phase elements (CPEs). The numerical values of fitted charge transfer resistance are provided in Supplementary Table 6. Except for Ir film, the other three photoanodes have the lower value of $R_{ct,1}$ obtained from the first semicircle of Nyquist plots, indicating that the M-I-S structure enables the charges to easily transport from n-Si to the surface. However, the $R_{ct,1}$ of Ir film/NiO/Ni/ZrO₂/n-Si is two times higher than that of the others, mainly due to the resistance between NiO and Ir film. The $R_{ct,2}$ is the resistance corresponding to the interface between the surface of catalysts and electrolyte. The Ir SAs/NiO/Ni/ZrO₂/n-Si exhibits the lowest value of $R_{ct,2}$ among all the photoanodes, which implies that the fast photogenerated charge transfer kinetics is achieved at the active sites where Ir single atoms are anchored. From the IMPS and EIS measurement, the numerical values of $\eta_{trans}$ and $R_{ct, surface \to electrolyte}$ according to the morphology of Ir are plotted in Fig. 4f. The values of two parameters are completely inversely proportional and it proves the outstanding property of Ir single atoms to boost the photogenerated charge transport.

### Theoretical investigations on Ir SAs/NiO PEC catalysts

To further identify an atomic-level mechanism of OER activities on Ir SAs/NiO PEC catalysts, DFT calculations are carried out. The (100) surface of NiO is one of the most stable surfaces for OER and the atomic structure in which the Ir atoms occupied the Ni vacancies is adopted. The energetic pathway based on the 4e⁻ mechanism of alkaline OER on Ir SAs/NiO is demonstrated in Fig. 5a. For the comparison, bare NiO (100) and IrO₂ (110) were selected. In Fig. 5b, under U = 1.23 V versus RHE, the potential determining step (PDS) of Ir SAs/NiO (100) and IrO₂ (110) is the conversion of O* to OOH*, while that of NiO (100) is the oxidation of OH* to O*. Compared to the overpotential

of 1.09 and 0.633 V for NiO (100) and IrO₂ (110), respectively, the Ir SAs/NiO (100) exhibits the lowered thermodynamic energy barrier with the calculated theoretical overpotential of 0.621 V. From this result, it is identified that the atomically dispersed Ir catalysts on NiO matrix can serve as energetically favorable sites outperforming IrO₂. The free energy diagrams of OER at 0 V versus CHE are also provided in Supplementary Fig. 23. In addition, the reaction kinetics of Ir SAs/NiO (100) was examined to predict the kinetic barrier heights of the rate-determining step via climbing image nudged elastic band (CI-NEB) calculations in Supplementary Fig. 24. Calculated with the most favorable incident OH angle, a small kinetic barrier of about 0.05 eV is observed in the transition state where there is a repulsion between O* and OH. In Supplementary Fig. 25, the reaction energy profiles and energy barriers at other incident OH angles are provided. They show a slightly increased kinetic energy barriers because OH requires additional energy as it rotates for a favorable incident angle. In Fig. 5c–e, the charge density redistributions of Ir SAs/NiO, NiO, and IrO₂ combined with Bader charge analysis are demonstrated. The localized polarization between Ir single atoms and OH intermediates in Fig. 5c, comparable to the polarization at the surface of IrO₂ in Fig. 5e, is much larger than that between Ni atoms and OH in Fig. 5d. It suggests that a strong local electric field around the isolated Ir atom promotes the photoinduced hole transport to the adsorbate, facilitating the subsequent OER.

### Discussion

In summary, the Ir single atoms are successfully applied to a Si photoelectrode by anchoring on NiO/Ni thin-film catalysts through a single-cycle ALD process and the M-I-S structure is constructed by inserting the ZrO₂ layer. It is confirmed that the electronic state of the isolated Ir atoms is modulated due to the charge transfer between NiO and Ir species. Using the frequency-domain analysis such as IMPS and EIS, we observed the kinetics of the photogenerated charge carriers indicating that the atomically dispersed Ir catalysts enable the facile photogenerated charge transport at the surface by suppressing the

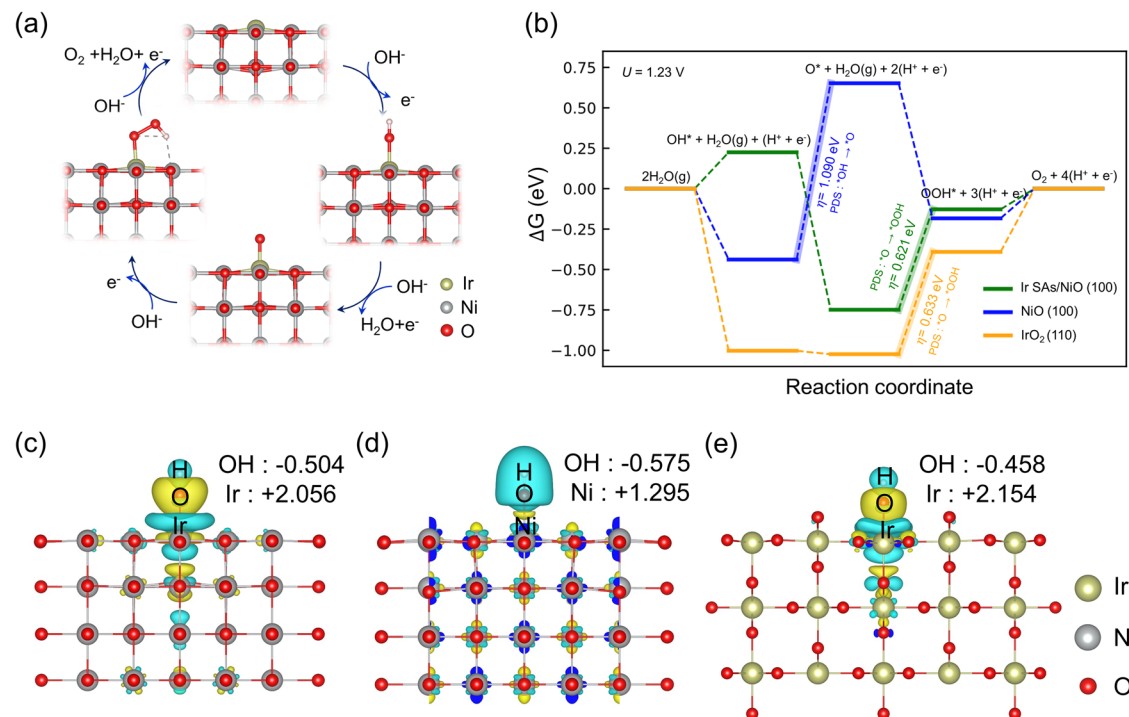

**Fig. 5 | DFT calculations. a** Proposed reaction pathway of oxygen evolution reaction on Ir SAs/NiO. **b** Free energy diagrams of OER on Ir SAs/NiO (100), NiO (100), and IrO₂ (110) at 1.23 V vs. RHE. Charge transfer diagrams of **c** Ir SAs/NiO, **d** NiO, **e** IrO₂ during OER.

charge recombination. Also, DFT calculations revealed that the isolated Ir atoms lower the thermodynamic energy barrier of the conversion from O* to OOH* and make the photoinduced holes easily transfer to the adsorbates. Consequently, Ir SAs/NiO/Ni/ZrO₂/n-Si exhibits a record-high PEC-OER performance with the photocurrent density of 27.7 mA cm⁻² at 1.23 V versus RHE and 130 h stability. This work broadens our fundamental understanding of the role of the single atoms in PEC application and proposes a novel and effective strategy to construct atomic-scale architecture on Si photoelectrode.

## Methods

### Fabrication of Ir SAs & NCs & film/NiO/Ni/ZrO₂/n-Si

The phosphorus-doped (100) n-type silicon wafers (1–10 Ω cm) were cut into pieces with a size of $1.5 \times 1.5$ cm². The pieces of silicon wafers were cleaned with acetone, isopropyl alcohol, and DI water by ultrasonication for 15 min at each step and soaked into HF for 10 s to remove the $SiO_2$ layer. The $ZrO_2$ layer with 1 nm thickness and Ni layer with 2 nm were deposited on Si wafer sequentially by electron beam evaporator. The growth rate and pressure of the chamber were 0.1 Å s⁻¹ and $1.0 \times 10^{-6}$ mTorr, respectively.

Ir-ALD was performed in a traveling-wave type ALD reactor (NCD Technology, Lucida D100, Korea) using tricarbonyl (1,2,3-η)-1,2,3-tri(tert-butyl)-cyclopropenyl iridium ($C_{18}H_{27}IrO_3$ or TICP) (Tanaka Precious Metals, Japan) and $O_2$ (99.99%). The temperature during the deposition was 250 °C. An optimized vapor pressure during deposition was obtained by heating the precursor at 55 °C. As the carrier and purge gas for the metal precursor, 100 sccm of $N_2$ flowed. A standard cycle of the Ir deposition process includes precursor pulsing for 7 s, precursor purging for 10 s, reactant pulsing for 5 s, and reactant purging for 10 s. For the synthesis of Ir single atoms, only one cycle of ALD has proceeded. For the synthesis of Ir nanoclusters, film, and film-T, 25, 100, and 200 cycles of ALD were proceeded, respectively. The NiO/Ni/ZrO₂/n-Si is fabricated by thermally annealing Ni/ZrO₂/n-Si with the same condition of the ALD process without Ir precursor.

### Characterizations

The TEM specimen of Ir SAs/NiO/Ni/ZrO₂/Si sample for the cross-sectional image was prepared by focused ion beam (FIB, SMI3050SE, SII Nanotechnology). The transparent FIB-prepared specimen was analyzed by TEM (JEM-2100F, JEOL) to view the sample in a direction parallel to the surface. The aberration-corrected high-angle annular dark field-scanning TEM images and energy dispersive X-ray (EDX) mappings were obtained with Cs corrected monochromated TEM (Themis Z 60-300, Thermofisher). To obtain the catalyst layer on the Cu TEM grid that is as similar to the sample as possible, all the layers (Ir SAs, NiO/Ni, and ZrO₂) were deposited on the Cu TEM grid in the same manner as deposition on the sample. The operation voltage was 300 kV. The semiangle of the probe-forming aperture was 17.9 mrad. The inner and outer semiangles of the HAADF detector were ~50 and 200 mrad. The probe current and dwelling time were 57.9 pA and 2 μs. The surface morphology was identified by FE-SEM (MERLIN COMPACT, JEISS). Using XPS (AXIS SUPRA, Kratos), the chemical composition and electronic states were analyzed. The background of XPS data was corrected by the Shirley method and the peaks were fitted using XPSpeak41 software. The XAFS data at the Ir L₃-edge was acquired on beamline 8C at the Pohang Light Source (PLS) in the Pohang Accelerator Laboratory (PAL), Republic of Korea. The data were obtained in the fluorescence mode by a solid-state detector. The Athena and Artemis in Demeter software were utilized to transform and process the data. X-ray diffractometer (D8 discover, Bruker) with Cu Kα radiation was performed to confirm the crystal structure and crystallinity. The concentration and weight percent of Ir were obtained by ICP-MS (NexION 350D, Perkin-Elmer). The transmittance versus wavelength was measured by UV-visible spectroscopy (V-770, JASCO).

## Photoelectrochemical measurements

A Xe arc lamp (Abet Technologies, LS150) was used as a light source and its intensity was calibrated to 1 sun (100 mW cm⁻², AM 1.5 G) with a reference diode. The PEC measurements were carried out using a potentiostat (Ivium Technologies, Nstat) with a three-electrode system using a saturated Ag/AgCl, Pt plate, and 1 M NaOH for reference electrode, counter electrode, and electrolyte, respectively. In acidic condition, a saturated calomel electrode (SCE), graphite rod, and 0.5 M $H_2SO_4$ were used. For LSVs, the potential was swept toward the anodic direction at a scan rate of 10 mV s⁻¹. All the potentials are converted into RHE according to the Nernst equation.

$$E(RHE) = E(Ag/AgCl) + E^0(Ag/AgCl) + 0.059 \times pH \qquad (6)$$

$$E(RHE) = E(SCE) + E^0(SCE) + 0.059 \times pH \qquad (7)$$

E(Ag/AgCl) is the measured potential versus the Ag/AgCl reference electrode through a potentiostat and $E^0$(Ag/AgCl) is 0.198 V at 25 °C. E(SCE) is the measured potential versus the SCE reference electrode through a potentiostat and $E^0$(SCE) is 0.241 V at 25 °C. Electrochemical impedance spectroscopy (EIS) was conducted with a frequency range from 250 kHz to 1 Hz and an amplitude of 10 mV. The measured EIS data were fitted to the equivalent circuits using Z plot software. The incident-photon-to-current conversion efficiency (IPCE) was carried out using a monochromator (MonoRa150) with an applied potential of 1.23 V versus RHE. The long-term stability data was obtained by chronoamperometry at the potential of 1.23 V versus RHE. To measure the amount of the evolved $O_2$ and calculate the Faradaic efficiency, the gas chromatography system (Agilent GC 7890B) was adopted. For intensity-modulated photocurrent spectroscopy (IMPS) measurements, an electrochemical workstation (Zennium, Zahner) and a potentiostat (PP211, Zahner) were utilized. The periodic modulation was swept with the frequency from 100 kHz to 0.1 Hz. The white light with 300 W m⁻² intensity was illuminated. The $k_{trans}$ and $k_{rec}$ can be induced by Eqs. (8) and (9).

$$k_{trans} + k_{rec} = 2\pi f \qquad (8)$$

$$\text{Charge transfer efficiency} = k_{trans}/(k_{trans} + k_{rec}) \qquad (9)$$

$f$ is the frequency at which the imaginary photocurrent is at its maximum and charge transfer efficiency can be obtained by the ratio of real photocurrents at the low-frequency and high-frequency intercepts.

## Computational details

In this study, all the spin-polarized density functional theory (DFT) calculations were carried out using the Vienna Ab-initio Simulation Package (VASP) with the projector augmented wave method for the core region and a plane-wave kinetic energy cutoff of 450 eV[55–57]. The generalized gradient approximation (GGA) in the form of Perdew-Burke-Ernzerhof (PBE) for the exchange-correlation potentials was used. The DFT + U calculations were performed with Hubbard-U correction of U = 6.45 eV to the d-electrons of Ni to account for the on-site correlation effects. The optimized lattice constants of NiO (rock-salt) and $IrO_2$ (rutile) are a = 4.180 Å, and a = 4.564 Å, c = 3.200 Å, respectively. One Ni atom at the outermost layer of NiO (001) surface was replaced with an Ir atom in order to simulate Ir single-atom on NiO (001). The oxygen evolution reaction (OER) on the surfaces of NiO (001) and $IrO_2$ (110) is carried out using the slab models composed of p(3 × 3) supercells with four and twelve atomic monolayers, respectively. The large vacuum layers of these slab models were set at least 15 Å in the z direction for the isolation of the surface to prevent the interaction between two periodic units. A 4 × 4 × 1 Gamma-centered Monkhorst-pack sampled k-point

grid was employed to sample the reciprocal space for the slab models. The bottom two atomic monolayers of NiO (001) and three atomic monolayers of $IrO_2$ (110) are fixed at their bulk positions while the other atomic layers and adsorbates are free to move in all directions until the convergence of energy and residual force on each atom were less than $1 \times 10^{-5}$ eV and 0.05 eV/Å, respectively.

According to the OER cycle proposed by Nørskov, the OER reaction follows the four-electron mechanism, corresponding to the four primitive steps listed, which involve adsorbed OH, O, and OOH intermediates on the surface (*),

$$* + OH^- \rightarrow {}^*OH + e^- \tag{10}$$

$$^*OH + OH^- \rightarrow H_2O + {}^*O + e^- \tag{11}$$

$$^*O + OH^- \rightarrow {}^*OOH + e^- \tag{12}$$

$$^*OOH + OH^- \rightarrow O_2 + H_2O + e^- \tag{13}$$

For the reaction step containing the coupled proton-electron pair, computational hydrogen electrode (CHE) model was used to obtain the free energy of a pair of proton and electron as half of the free energy of hydrogen gas molecule at conditions with U = 0 V and $P_{H2} = 1$ bar. The chemical potential of each adsorbate is determined as $\mu = E + E_{ZPE} - T \times S$, where $E$ is the total energy acquired from DFT calculations, $E_{ZPE}$ is zero-point energy and $S$ is the entropy at 298 K. The reaction free energy of each elementary step of OER was obtained by the adsorption free energy calculations of O*, OH*, OOH*, and H*. The adsorption free energy $\Delta G_{O^*}$, $\Delta G_{OH^*}$, and $\Delta G_{OOH^*}$ in relation to the free energy of stoichiometrically appropriate amounts of $H_2O$ (g) and $H_2$ (g) were defined as follows:

$$\Delta G O_* = \mu O_* + \mu_{H2} - \mu_{H2O} - \mu_* \tag{14}$$

$$\Delta_{GOH^*} = \mu_{OH}^* + 0.5 \times \mu_{H2} - \mu_{H2O} - \mu_* \tag{15}$$

$$\Delta G_{OOH^*} = \mu_{OOH^*} + 1.5 \times \mu_{H2} - 2 \times \mu_{H2O} - \mu^* \tag{16}$$

The reaction free energy of equations for OER were calculated using the following equations:

$$\Delta G_1 = \Delta G_{OH^*} \tag{17}$$

$$\Delta G_2 = \Delta G_{O^*} - \Delta G_{OH^*} \tag{18}$$

$$\Delta G_3 = \Delta G_{OOH^*} - \Delta G_{O^*} \tag{19}$$

$$\Delta G_4 = 4.92 - \Delta G_{OOH^*} \tag{20}$$

Under ideal conditions, the OER reaction with a total energy change of 4.92 eV can be driven at 1.23 V, while the $\Delta G$ of each elementary reaction would be equally divided into 1.23 eV. Therefore, the theoretical overpotential $\eta$ was introduced to represent additional required potential and measure the catalytic activity of materials, which was defined in theoretical calculations as,

$$\eta = \max[\Delta G_1, \Delta G_2, \Delta G_3, \Delta G_4]/e - 1.23V \tag{21}$$

The potential determining step (PDS) is determined as the final step to become downhill in free energy with the increase of potential.

To predict kinetic barriers of the transition state at the potential determining step, the climbing image nudged elastic band method was adopted[58]. Damped molecular dynamics and quick-min force-based optimizer were used and the self-consistent calculations of single-electron wavefunction were terminated when the iterative convergence of energy and force fulfilled $10^{-4}$ eV and 0.1 eV/Å, respectively.

### Reporting summary
Further information on research design is available in the Nature Portfolio Reporting Summary linked to this article.

## Data availability
The data that support the findings of this study are available from the corresponding authors upon reasonable request. Source data are provided with this paper.

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

## Acknowledgements

This research was supported by National Research Foundation of Korea (NRF) grant funded by the Korea governmnet Ministry of Science and ICT (MSIT) (2021R1A4A4A3027878, 2018M3D1A1058793, and

2021M3H4A1A03057403). This research was also supported by the KRISS (Korea Research Institute of Standards and Science) MPI Lab. program. H. W. Jang and S. E. Jun gratefully acknowledge KRISS MPI Lab. program. The Inter-University Semiconductor Research Center and Institute of Engineering Research at Seoul National University provided research facilities for this work. We thank the Pohang Accelerator Laboratory for providing the synchrotron radiation source at 8C beamline.

## Author contributions

H.W.J., S.H.K., and J.M. supervised the project. H.W.J. and S.E.J. conceived the project. S.E.J. fabricated and measured the devices and analyzed the experimental results. Y.H.K. carried out ALD operation. J.H.K. performed density first theory calculations. W.S.C. conducted gas chromatography measurements. S.C. carried out TEM characterization. J.Y. helped to measure UV-visible spectroscopy. H.P., S.H.P., and K.C.K. helped to analyze the experimental results. H.L. helped to measure and analyze IMPS. The manuscript was mainly written by S.E.J. and H.W.J. All authors discussed the results and commented on the manuscript at all stages.

## Competing interests

The authors declare no competing interests.
