## [Peer Review File · Nature Communications]

REVIEWER COMMENTS

Reviewer #1 (Remarks to the Author):

Review for NCOMMS-22-17185

In this article, Jun et al. report a Si-based photoanode made of several coatings: a ZrO₂ tunnel layer, a Ni/NiO collector, and Ir single atoms. The photoanode presents state-of-the-art performance for an n-Si photoelectrode in the frame of oxygen evolution reaction at high pH and the approach is original because the use of single atoms in this kind of device has not been yet explored. Overall, the manuscript is well-written, the characterization is thorough and simulation has been performed to understand the observed effect. However, some important aspects are missing. I think that the manuscript has the potential to be published in Nature Communications if the following points are addressed:

-Are the TEM images of Figure 1 cross-sections cut perpendicular to the surface or in a parallel fashion? The preparation procedure is not described. If the TEM images of Figure 2 are cross-sections, why the Ir atoms are present within the whole layer and observed deep inside the Ni/NiO matrix? ALD is used to deposit Ir over Ni, it is expected that Ir should only be present on the outer layer.

-Important: it is not clear what is the effect of the NiO and that of Ir single atoms? The authors should add the voltammogram of NiO/Ni/ZrO₂/n-Si to Figure 3a. If possible, an Ir(SA)/ZrO₂/n-Si can also be added.

-Important: Photovoltage is not determined, it is required that the authors measure the voltammogram of a degenerate anode (using highly doped p⁺⁺Si) prepared the same way as the best photoanode, i.e., Ir(SA)/NiO/Ni/ZrO₂/p⁺⁺Si. This measurement can be used to precisely determine the photovoltage

-Does Ir affect the photovoltage, or just plays a role in collecting minority carriers and catalysis?

-Important: Post-utilization analyses and particularly XPS is required to know how Ir atoms evolve during operation

Reviewer #2 (Remarks to the Author):

Jun. et al in this paper explored a single atom Ir decorated Si photoanode for efficient water oxidation. Firstly, the preparation procedure and material interface characterization are demonstrated. Further, photoelectrochemical OER activity was investigated. Lastly, the interface charge carrier kinetics was analyzed by frequency-domain analysis, which provides critical information related to the role of single atom Ir. Though the manuscript is written in a comprehensive manner, there are some fundamental issues that need to be addressed.

1. One main issue with this work is that all the component is well studied, and the interface is a combination of a well-studied system that influence the novelty of this work. For instance, Ni/NiO has been demonstrated to be great protection and catalytic interface for n-Si (Science, 2013, 342, 836-840, ACS Catalysis. 2018, 8,7261-7269). Ir single atom has been demonstrated to be an excellent catalyst for oxide support for water oxidation (Nature communication, 2022, 13, 24; PNAS, 2018, 115, 2902-2907).

2. A large number of experimental details are missing or unclear. For instance, in order to confirm the existence of Ir single atom on top of Si photoelectrode. High-resolution TEM images are taken (Fig. 1). However, the photoelectrode's sample is not transparent, how the TEM sample is prepared and how the Ir single atom is identified is unrecorded.

3. The chemical states of Ir single atom are unclear. For instance, based on XPS results, it is hard to distinguish what is the oxidation state of Ir since its peak is tiny. Further, the Ir SAs XANES spectra demonstrate it is very similar to IrO₂. Based on the literature, the Ir single atom's oxidation state should be very different from IrO₂, slightly smaller than +3(Nature communication, 2022, 13, 24). Would it be possible that Ir is oxidized to IrO₂ in the ALD process?

4. The author claim that the existence of the ZrO₂ layer between n-Si and Ni reduces the surface states and eliminated the Fermi level pinning effect (Fig. 1S). However, there is no control sample or experimental evidence supporting this claim.

5. Why there is an IPCE efficiency decrease at 630 nm? A control sample of IPCE without Ir SAs, without Ni layer, and without ZrO₂, should be provided.

6. The IMPS study is a great addition to this paper. However, the author should comment on the circuit model from the fitting, to if it is valid and suitable for a complicated interface like the one demonstrated in this paper. Since based on different systems, the circuit model can vary a lot (J. Phys. Chem. C. 2019, 123, 41, 24995-25014).

Response to the Reviewers' Comments

We thank the reviewers for their constructive comments and suggestions that are helpful for us to improve the manuscript, entitled “*Atomically Dispersed Ir Catalysts on Si Photoanode for Efficient Photoelectrochemical Water Splitting*” (NCOMMS-22-17185). We have fully revised the manuscript taking into account of the reviewer’s comments. A point-by-point response to the reviewers’ comments is given below.

Reviewer #1:

In this article, Jun et al. report a Si-based photoanode made of several coatings: a ZrO₂ tunnel layer, a Ni/NiO collector, and Ir single atoms. The photoanode presents state-of-the-art performance for an n-Si photoelectrode in the frame of oxygen evolution reaction at high pH and the approach is original because the use of single atoms in this kind of device has not been yet explored. Overall, the manuscript is well-written, the characterization is thorough and simulation has been performed to understand the observed effect. However, some important aspects are missing. I think that the manuscript has the potential to be published in Nature Communications if the following points are addressed:

Response: We are grateful for the time and effort that Reviewer 1 has spent in reviewing our manuscript. Please see our point-by-point responses to the reviewer’s comment.

1. Are the TEM images of Figure 1 cross-sections cut perpendicular to the surface or in a parallel fashion? The preparation procedure is not described. If the TEM images of Figure 2 are cross-sections, why the Ir atoms are present within the whole layer and observed deep inside the Ni/NiO matrix? ALD is used to deposit Ir over Ni, it is expected that Ir should only be present on the outer layer.

We appreciate the reviewer for the kind and thoughtful comment. Fig. 1a is an image of the sample cut in a direction perpendicular to the surface and viewed in a direction parallel to the surface. The preparation procedure of the sample is as follows: The TEM specimen of Ir SAs/NiO/Ni/ZrO₂/Si sample for the cross-sectional image was prepared by focused ion beam

(FIB, SMI3050SE, SII Nanotechnology). The transparent FIB-prepared specimen was analyzed by TEM (JEM-2100F, JEOL) to view the sample in a direction parallel to the surface.

Just like the reviewer thought, atomically dispersed Ir elements exist only on the surface of NiO and not deep in the NiO/Ni matrix. Fig. 1b and c are top-view TEM images where only the surface of NiO can be seen. We presented the images to identify the atomic structure of NiO and how the Ir SAs are distributed on the surface of NiO. To clarify the view-point of the images, we revised the caption regarding Fig. 1b and c.

→ According to this comment, we have revised the manuscript.

“The TEM specimen of Ir SAs/NiO/Ni/ZrO₂/Si sample for the cross-sectional image was prepared by focused ion beam (FIB, SMI3050SE, SII Nanotechnology). The transparent FIB-prepared specimen was analyzed by TEM (JEM-2100F, JEOL) to view the sample in a direction parallel to the surface.”

“**Fig. 1 Transmission electron microscope analysis of Ir SAs/NiO/Ni/ZrO₂/n-Si.** **a** Cross-sectional TEM image of Ir SAs/NiO/Ni/ZrO₂/n-Si photoanodes. **b** High-resolution top-view TEM image of Ir SAs/NiO surface and corresponding fast Fourier transformation (FFT) pattern. **c** Top-view HAADF-STEM image of Ir SAs/NiO thin-film catalyst. **d, e** Intensity profiles (line 1 and 2) showing the intensity difference between Ni and Ir atoms. **f** STEM-EDS elemental mapping of Ir SAs/NiO thin-film.”

2. Important: it is not clear what is the effect of the NiO and that of Ir single atoms? The authors should add the voltammogram of NiO/Ni/ZrO₂/n-Si to Figure 3a. If possible, an Ir(SA)/ZrO₂/n-Si can also be added.

We appreciate the reviewer for your helpful comment. In photoelectrochemistry, NiO acts as both a catalyst and passivation layer to boost the oxygen evolution reaction and protect the photoelectrode. Also, it is optically transparent and has an index of refraction which enables it to become an anti-reflective coating. In this electrode, NiO not only plays these roles but also serves to anchor Ir single atoms. The anchored Ir single atoms with a modified electronic state boost the PEC performance by enabling the facile photogenerated charge transport at the surface by suppressing the charge recombination.

As the reviewer suggested, we fabricated NiO/Ni/ZrO₂/n-Si photoanode by annealing Ni/ZrO₂/n-Si with the same condition of the ALD process without Ir precursor and added the voltammograms in Figure 3a to distinguish the effect of NiO and Ir SAs. From the lowered onset potential and increased saturation current compared to Ni/ZrO₂/n-Si, it is confirmed that NiO enhanced catalytic activity and photon absorption of silicon photoanode.

Furthermore, in addition to the reviewer's recommendation, the LSV curves of NiO/Ni/ZrO₂/n-Si, Ir ALD-1cyc./ZrO₂/n-Si, and Ir SAs/NiO/Ni/n-Si are obtained to verify the effect of each layer. For the electrode without NiO/Ni, Ir atoms are not anchored on the surface, resulting in no photoelectrochemical catalytic activity. From this result, we can ensure the role of NiO/Ni to anchor Ir SAs on its lattice. Moreover, the role of ZrO₂ increasing the photovoltage can be seen through the photoanode where ZrO₂ does not exist.

→ According to this comment, we have revised the manuscript, Fig. 3a, and Supplementary Fig. 11.

In manuscript,

“Linear sweep voltammograms (LSVs) of Ni/n-Si, Ni/ZrO₂/n-Si, NiO/Ni/ZrO₂/n-Si, and Ir SAs/NiO/Ni/ZrO₂/n-Si photoanodes are shown in Figure 3a.”

“The thermally oxidized Ni surface in NiO/Ni/ZrO₂/n-Si induces enhanced catalytic activity and photon absorption, which is confirmed by lowered onset potential and increased saturation current. The Ir SAs/NiO/Ni/ZrO₂/n-Si photoanode exhibits dramatically enhanced photoelectrochemical performance with the onset potential of 0.97 V vs. RHE and the current density of 27.7 mA cm⁻² at 1.23 V vs. RHE. It is attributed to the capability of Ir SAs that enable the facile charge transport by suppressing the charge recombination and boost the catalytic surface reaction. To verify the role of each layers, the LSV curves of NiO/Ni/ZrO₂/n-Si, Ir ALD-1cyc./ZrO₂/n-Si, and Ir SAs/NiO/Ni/n-Si are provided in Supplementary Fig. 11. For the photoelectrode without NiO/Ni, Ir atoms are not anchored on the surface, resulting in no photoelectrochemical catalytic activity. Also, the role of ZrO₂ increasing the photovoltage can be seen through the photoanode where ZrO₂ does not exist.”

“For the synthesis of Ir nanoclusters, film, and film-T, 25, 100, and 200 cycles of ALD were proceeded, respectively. The NiO/Ni/ZrO₂/n-Si is fabricated by thermally annealing Ni/ZrO₂/n-Si with the same condition of the ALD process without Ir precursor.”

In Fig. 3,

a Linear sweep voltammograms (LSVs) of the Ni/n-Si, Ni/ZrO₂/n-Si, NiO/Ni/ZrO₂/n-Si, and Ir SAs/NiO/Ni/ZrO₂/n-Si photoanodes measured in 1 M NaOH electrolyte.

In supplementary information,

Figure S11. LSVs of NiO/Ni/ZrO₂/n-Si, Ir ALD-1cyc./ZrO₂/n-Si, and Ir SAs/NiO/Ni/n-Si.

3. Important: Photovoltage is not determined, it is required that the authors measure the voltammogram of a degenerate anode (using highly doped p⁺⁺Si) prepared the same way as the best photoanode, i.e., Ir(SA)/NiO/Ni/ZrO₂/p⁺⁺Si. This measurement can be used to precisely determine the photovoltage.

We thank the reviewer for the insightful suggestion. To precisely determine the photovoltage of the device, we prepared the Ir SAs/NiO/Ni/ZrO₂/p⁺⁺-Si anode and obtained the linear sweep voltammogram in the same measurement condition with Ir SAs/NiO/Ni/ZrO₂/n-Si photoanode. When comparing the onset potential of each LSV curve, it is confirmed that the photovoltage of the photoanode was 550 mV.

→ According to this comment, we have revised the manuscript and Supplementary Fig. 15.

In manuscript,

“For the electrochemical (EC) measurements in Supplementary Fig. 11, the same tendency is shown in catalytic activity. To precisely determine the photovoltage, Ir SAs/NiO/Ni/ZrO₂ is deposited on highly doped p⁺⁺-Si and the LSV curve is obtained in Supplementary Fig. 15. When comparing the onset potential of Ir SAs/NiO/Ni/ZrO₂/n-Si and Ir SAs/NiO/Ni/ZrO₂/p⁺⁺-Si, it is confirmed that the photovoltage of the photoanode is 550 mV.”

In supplementary information,

Figure S15. LSV curves of Ir SAs/NiO/Ni/ZrO₂/n-Si photoanode and Ir SAs/NiO/Ni/ZrO₂/p⁺⁺-Si anode for determining the photovoltage of the device.

4. Does Ir affect the photovoltage, or just plays a role in collecting minority carriers and catalysis?

We appreciate the reviewer for your valuable comment. In photoelectrochemistry, the photovoltage of the semiconductor is determined by the quasi-Fermi level difference at the interfaces under light illumination. To enhance the photovoltage of the photoelectrode, various strategies have been implemented such as passivation of surface states, band alignment engineering, and so on. In this work, however, Ir catalyst does not affect the photovoltage of the device as it does not change the surface states of Si photoanode and band alignment. It collects the photogenerated holes by suppressing the charge recombination and boosts the catalytic surface reaction. To clarify the role of Ir SAs, we commented on the reason why the PEC performance increased after Ir SAs have been anchored.

→ According to this comment, we have revised the manuscript.

“The Ir SAs/NiO/Ni/ZrO₂/n-Si photoanode exhibits dramatically enhanced photoelectrochemical performance with the onset potential of 0.97 V vs. RHE and the current density of 27.7 mA cm⁻² at 1.23 V vs. RHE. It is attributed to the capability of Ir SAs that

enable the facile charge transport by suppressing the charge recombination and boost the catalytic surface reaction.”

5. Important: Post-utilization analyses and particularly XPS is required to know how Ir atoms evolve during operation.

We appreciate the constructive suggestion from the reviewer. We do agree with the importance of analyzing post-utilized samples. Thus, XPS analysis was conducted with Ir SAs/NiO/Ni/ZrO₂/n-Si sample after 130 h stability test. In chronoamperometry after 130 h, the performance of the device gradually degraded due to the detachment of both NiO/Ni layer and Ir single atoms. In XPS, the peaks of Ni 3p and Ir 4f disappeared while Na 2s peak appeared at 63.1 eV derived from the surface poisoning of NaOH electrolyte.

→ According to this comment, we have revised the manuscript and Supplementary Fig. 17.

In manuscript,

“It demonstrates the remarkably stable PEC performance with 130 h, which is an encouraging result considering that Si photoelectrode is highly vulnerable to an alkaline environment. It is attributed to a chemically robust NiO/Ni catalyst and its capability to stabilize and activate Ir single atoms through strong interactions. However, after 130 h stability test, the performance gradually degraded due to the detachment of the catalyst layer and surface damage by the NaOH electrolyte (Supplementary Fig. 17).”

In supplementary information,

Figure S17. a Chronoamperometry showing the degradation after 130 h operation. **b** Na 2s spectra of the device after stability test.

Reviewer #2:

Jun. et al in this paper explored a single atom Ir decorated Si photoanode for efficient water oxidation. Firstly, the preparation procedure and material interface characterization are demonstrated. Further, photoelectrochemical OER activity was investigated. Lastly, the interface charge carrier kinetics was analyzed by frequency-domain analysis, which provides critical information related to the role of single atom Ir. Though the manuscript is written in a comprehensive manner, there are some fundamental issues that need to be addressed.

Response: We are grateful for the time and effort that Reviewer 2 has spent in reviewing our manuscript. Please see our point-by-point responses to the reviewer's comment.

1. One main issue with this work is that all the component is well studied, and the interface is a combination of a well-studied system that influence the novelty of this work. For instance, Ni/NiO has been demonstrated to be great protection and catalytic interface for n-Si (Science, 2013, 342, 836-840, ACS Catalysis. 2018, 8,7261-7269). Ir single atom has been demonstrated to be an excellent catalyst for oxide support for water oxidation (Nature communication, 2022, 13, 24; PNAS, 2018, 115, 2902-2907).

We sincerely thank the reviewer for the valuable and insightful comment. The novelty of this work lies in <1> the electronic structure and environmental coordination of Ir single atoms anchored on extremely thin film metal oxide optimized for PEC application and <2> the photogenerated charge carrier dynamics around Ir single atoms. In PEC application, it is necessary to synthesize thin film catalysts with sub-nanometer thickness to maximize the light absorption. However, depositing single atoms on such a thin film has not been attempted. In addition, it is the first time to apply single atoms into silicon photoelectrode and analyze them using TEM and XAFS. Moreover, most previous reports have focused on the catalytic effect of single atoms, but we first analyzed the kinetics of photogenerated charge carriers around single atoms. It is revealed that atomically dispersed Ir catalysts enable the facile photogenerated charge transport by suppressing the charge recombination. It would be greatly appreciated if the reviewer could focus on the novelty we were thinking of.

2. A large number of experimental details are missing or unclear. For instance, in order to confirm the existence of Ir single atom on top of Si photoelectrode. High-resolution TEM images are taken (Fig. 1). However, the photoelectrode's sample is not transparent, how the TEM sample is prepared and how the Ir single atom is identified is unrecorded.

We appreciate the reviewer for carefully reviewing our manuscript and providing the kind comment. As the reviewer mentioned, the sample itself is not transparent to obtain TEM images. For the best alternative to acquire the catalyst layer on the Cu TEM grid that is as similar to the sample as possible, all the layers (Ir SAs, NiO/Ni, and ZrO₂) were deposited on the Cu TEM grid in the same manner as deposition on the device. For the detailed explanation, we described the method for the preparation of TEM samples in Characterization. Furthermore, we additionally explained the experimental details regarding the procedure of HAADF-STEM, FIB, and XAFS.

→According to this comment, we have revised the manuscript.

“The aberration-corrected high-angle annular dark field-scanning TEM images and energy dispersive X-ray (EDX) mappings were obtained with Cs corrected monochromated TEM (Themis Z 60-300, Thermofisher). To obtain the catalyst layer on the Cu TEM grid that is as similar to the sample as possible, all the layers (Ir SAs, NiO/Ni, and ZrO₂) were deposited on the Cu TEM grid in the same manner as deposition on the sample.

“The operation voltage was 300 kV. The semiangle of the probe-forming aperture was 17.9 mrad. The inner and outer semiangles of the HAADF detector were ~50 and 200 mrad. The probe current and dwelling time were 57.9 pA and 2 μs.”

“The TEM specimen of Ir SAs/NiO/Ni/ZrO₂/Si sample for the cross-sectional image was prepared by focused ion beam (FIB, SMI3050SE, SII Nanotechnology). The transparent FIB-prepared specimen was analyzed by TEM (JEM-2100F, JEOL) to view the sample in a direction parallel to the surface.”

“The data were obtained in the fluorescence mode by a solid-state detector. The Athena and Artemis in Demeter software were utilized to transform and process the data.”

3. The chemical states of Ir single atom are unclear. For instance, based on XPS results, it is hard to distinguish what is the oxidation state of Ir since its peak is tiny. Further, the Ir SAs XANES spectra demonstrate it is very similar to IrO₂. Based on the literature, the Ir single atom's oxidation state should be very different from IrO₂, slightly smaller than +3(Nature communication, 2022, 13, 24). Would it be possible that Ir is oxidized to IrO₂ in the ALD process?

We appreciate the reviewer for your helpful comment. According to the reviewer's advice, the magnified and simplified version of Fig. 2a was provided in Supplementary Fig. 7 for better visualization and clarification of XPS results. With this figure, the oxidation state of Ir SAs can be clearly identified. It indicates that atomically dispersed Ir atoms on NiO lattice exist mainly at the +3 ~ +4 valence states.

The straightlines for white line peaks in XANES spectra were unclear to compare the oxidation state of Ir element. To this end, we newly marked the straightlines to make them stand out. In this figure, the white line peak position of Ir SAs is located between that of IrO₂ and metallic Ir, implying the existence of oxidation state between +4 and zero-valence. However, it is located closer to IrO₂, and from this, it can be seen that the oxidation state of Ir SAs is not +4,

but close to it. We kindly refer to the references in which the oxidation state of single atoms is very similar to that of metal oxide when single atoms are deposited by ALD (J. Am. Chem. Soc. 2019, 141, 14515-14519, Angew. Chem. Int. Ed. 2018, 57, 909-913).

→ According to this comment, we have revised the manuscript, Fig. 2b, and Supplementary Fig. 7.

In manuscript,

“For better visualization and clarification of XPS results, the magnified and simplified version of Fig. 2a is provided in Supplementary Fig. 7.”

In Fig. 2,

In supplementary information,

Figure S7. The magnified and simplified Ir 4f spectra of NiO/Ni/ZrO₂/n-Si photoanodes with Ir SAs, NCs, and film.

4. The author claim that the existence of the ZrO₂ layer between n-Si and Ni reduces the surface states and eliminated the Fermi level pinning effect (Fig. 1S). However, there is no control sample or experimental evidence supporting this claim.

Thanks a lot for your fruitful comment. We explained that the ZrO₂ layer reduced the interface states of Si and eliminated the Fermi level pinning effect between n-Si and Ni. Due to the enhanced photovoltage derived from large band bending, the LSV curve of Ni/ZrO₂/n-Si photoanode shifted to the cathodic direction compared to that of Ni/n-Si in Fig. 3a. To further support it, we conducted a Mott-Schottky measurement to obtain the flat band potential (E_{fb}) of Ni/n-Si and Ni/ZrO₂/n-Si photoanodes. The negative shift in flat band potential of Ni/ZrO₂/n-Si (-0.7 V vs. Ag/AgCl) relative to that of Ni/n-Si (-0.4 V vs. Ag/AgCl) proves the enlarged photovoltage resulted from the elimination of Fermi level pinning effect.

→ According to this comment, we have revised the manuscript and Supplementary Fig. 9.

In manuscript,

“To achieve high photovoltage, the interfacial energetics are manipulated by applying the ZrO₂ tunneling oxide layer, leading to the formation of metal-insulator-semiconductor junctions. As a result, the onset potential shifts toward the cathodic direction with the value of 1.14 V versus RHE to reach 1 mA cm⁻². To further support it, Mott-Schottky measurements are conducted to acquire the flat band potential (E_{fb}) of Ni/n-Si and Ni/ZrO₂/n-Si photoanodes. The negative shift in flat band potential of Ni/ZrO₂/n-Si (-0.7 V vs. Ag/AgCl) relative to that of Ni/n-Si (-0.4 V vs. Ag/AgCl) proves the enlarged photovoltage resulted from the elimination of Fermi level pinning effect.”

In supplementary information,

Figure S9. Mott-Schottky plots for Ni/n-Si and Ni/ZrO₂/n-Si.

5. Why there is an IPCE efficiency decrease at 630 nm? A control sample of IPCE without Ir SAs, without Ni layer, and without ZrO₂, should be provided.

We appreciate the reviewer for your helpful comment and kind suggestion. During IPCE measurement at the wavelength from 400 to 800 nm, the light filter is changed at around 630 nm, resulting in the sharp decline of the graphs. However, it is possible to minimize this phenomenon by being careful to filter change when inputting reference data using a photodiode. As a result, we newly obtained the smooth IPCE data in Fig. 3d and the photoanodes showed the efficiency of 75, 71, and 40% on average.

As the reviewer recommended, we also acquired the LSV curves and IPCE data of the samples without Ir SAs, without NiO/Ni, and without ZrO₂ layer. The efficiencies of about 40, 65, and 0% were measured for the samples without Ir SAs, without NiO/Ni, and without ZrO₂ layer, respectively. In the case of the device without NiO/Ni, Ir atoms were not anchored on the surface, resulting in no photoelectrochemical catalytic activity.

→ According to this comment, we have revised the manuscript, Fig. 3d, Supplementary Fig. 11, and Supplementary Fig. 16.

In manuscript,

“For the photoanode with Ir film, it shows the efficiency of 40% on average over the entire visible light wavelength. The efficiency is significantly increased by introducing Ir nanoclusters, and it reaches up to 75% on average when the single atom Ir catalysts are anchored on the photoanode. The efficiencies of about 40, 65, and 0% are measured for NiO/Ni/ZrO₂/n-Si, Ir SAs/NiO/Ni/n-Si, and Ir ALD-1cyc./ZrO₂/n-Si, respectively, in Supplementary Fig. 16.”

In Figure 3,

In supplementary information,

Figure S11. LSVs of NiO/Ni/ZrO₂/n-Si, Ir ALD-1cyc./ZrO₂/n-Si, and Ir SAs/NiO/Ni/n-Si.

Figure S16. Incident-photon-to-current conversion efficiency of NiO/Ni/ZrO₂/n-Si, Ir ALD-1cyc./ZrO₂/n-Si, and Ir SAs/NiO/Ni/n-Si.

6. The IMPS study is a great addition to this paper. However, the author should comment on the circuit model from the fitting, to if it is valid and suitable for a complicated interface like the one demonstrated in this paper. Since based on different systems, the circuit model can vary a lot (J. Phys. Chem. C. 2019, 123, 41, 24995-25014).

We appreciate the reviewer for your constructive comment. As the reviewer mentioned, it is necessary to clarify if the circuit model is suitable for these devices having several interfaces. To this end, we present a revised energy band model of the charge transfer & recombination process and an equivalent circuit model for an illuminated photoanode/electrolyte contact with reference to this paper (J. Phys. Chem. C. 2019, 123, 41, 24995-25014).

A generalized theoretical energy band and equivalent circuit model of bare n-Si junctioned with electrolyte are represented in Fig. a and b (J. Electroanal. Chem., 1995, 396, 219-226). R_{sc} and C_{sc} are the space charge resistance and space charge capacitance, respectively. These factors are related to surface recombination of photogenerated minority carriers. R_{ct} and C_H are the charge transfer resistance and Helmholtz layer capacitance, respectively. From this model, the values of k_{trans} and k_{rec} were derived as the equation (1) and (2).

$$k_{trans} = \frac{1}{R_{ct}(C_{sc} + C_H)} \quad (1)$$

$$k_{rec} = \frac{1}{R_{sc}(C_{sc} + C_H)} \quad (2)$$

For the photoanodes in this study, the energy band diagram and equivalent circuit model are provided in Fig. c and d. There are three additional interfaces compared to bare n-Si, which are

ZrO₂-Ni/NiO, Ni/NiO-Ir, and Ir-electrolyte. The resistances and capacitances at each interface are represented by R₁ & C₁, R₂ & C₂, and R₃ & C₃. In terms of surface recombination, not only R_{sc} & C_{sc} but also R₁ & C₁, R₂ & C₂, and R₃ & C₃ contribute to the recombination of photogenerated holes from n-Si. As a result, they can be connected in series as can be seen in Fig. d.

However, from the EIS data, we can see that the resistance and capacitance of the interfaces up to the surface are much smaller than those of the interface being in contact with the electrolyte. Moreover, the resistance and capacitance of the interfaces up to the surface are all the same regardless of the samples. Therefore, if we assume that R_{sc} & C_{sc}, R₁ & C₁, and R₂ & C₂ are negligibly small, the circuit is simplified as can be seen in Fig. e. From this model, the values of k_{trans} and k_{rec} were derived as the equation (3) and (4).

$$k_{trans} = \frac{1}{R_{ct}(C_3 + C_H)} \quad (3)$$

$$k_{rec} = \frac{1}{R_3(C_3 + C_H)} \quad (4)$$

Consequently, the charge transfer and charge recombination constants are expressed by the resistances and capacitances at the Ir-electrolyte interface. Therefore, it is possible to compare k_{trans} and k_{rec} depending on the morphology of Ir catalysts.

Despite the assumption, we think that this circuit model and rate constant model for IMPS analysis are appropriate to compare the surface charge transfer and recombination depending on the morphology of Ir catalysts.

→ According to this comment, we have revised the manuscript and Supplementary Fig. 18.

In manuscript,

“Even though these equations are derived from a classical model based on a bare semiconductor being in contact with the electrolyte, they also can be applied into the photoelectrodes having numerous interfaces with the assumption⁵⁴ (Supplementary Fig. 18).”

In references,

“54. Ravishankar, S. et al. Intensity-modulated photocurrent spectroscopy and its application to perovskite solar cells. *J. Phys. Chem. C* **123**, 24995-25014 (2019).”

In supplementary information,

Figure S18. **a** A generalized theoretical energy band and **b** equivalent circuit model of bare n-Si being in contact with electrolyte for IMPS. **c** The energy band diagram and **d** equivalent

circuit model of Ir/NiO/Ni/ZrO₂/n-Si being in contact with electrolyte. e The simplified circuit model with the assumption.

A generalized theoretical energy band and equivalent circuit model of bare n-Si junctioned with electrolyte are represented in Supplementary Fig. 18a and b¹. R_{sc} and C_{sc} are the space charge resistance and space charge capacitance, respectively. These factors are related to surface recombination of photogenerated minority carriers. R_{ct} and C_H are the charge transfer resistance and Helmholtz layer capacitance, respectively. From this model, the values of k_{trans} and k_{rec} were derived as the equation (1) and (2).

$$k_{trans} = \frac{1}{R_{ct}(C_{sc} + C_H)} \quad (1)$$

$$k_{rec} = \frac{1}{R_{sc}(C_{sc} + C_H)} \quad (2)$$

For the photoanodes in this study, the energy band diagram and equivalent circuit model are provided in Supplementary Fig. 18c and d. There are three additional interfaces compared to bare n-Si, which are ZrO₂-Ni/NiO, Ni/NiO-Ir, and Ir-electrolyte. The resistances and capacitances at each interface are represented by R₁ & C₁, R₂ & C₂, and R₃ & C₃. In terms of surface recombination, not only R_{sc} & C_{sc} but also R₁ & C₁, R₂ & C₂, and R₃ & C₃ contribute to the recombination of photogenerated holes from n-Si. As a result, they can be connected in series as can be seen in Supplementary Fig. 18d.

However, from the EIS data in Figure 4e, we can see that the resistance and capacitance of the interfaces up to the surface are much smaller than those of the interface being in contact with the electrolyte. Moreover, the resistance and capacitance of the interfaces up to the surface are all the same regardless of the samples. Therefore, if we assume that R_{sc} & C_{sc}, R₁ & C₁, and R₂ & C₂ are negligibly small, the circuit is simplified as can be seen in Supplementary Fig. 18e. From this model, the values of k_{trans} and k_{rec} were derived as the equation (3) and (4).

$$k_{trans} = \frac{1}{R_{ct}(C_3 + C_H)} \quad (3)$$

$$k_{rec} = \frac{1}{R_3(C_3 + C_H)} \quad (4)$$

Consequently, the charge transfer and charge recombination constants are expressed by the resistances and capacitances at the Ir-electrolyte interface. Therefore, it is possible to compare

k_{trans} and k_{rec} depending on the morphology of Ir catalysts regardless of the complicated interface.

REVIEWER COMMENTS

Reviewer #1 (Remarks to the Author):

The authors have answered the questions asked the reviewers and improved the manuscript, this paper should be now publishable in Nat Commun.

Reviewer #2 (Remarks to the Author):

The author has thoroughly answered the reviewers' previous questions. However, there are two questions that need to be further clarified.

1. In review 1 question 2, to confirm the role of ZrO₂, it will be great to compare the saturation current differences of Ni/n-Si and Ni/ZrO₂/n-Si. Unfortunately, the saturation current of Ni/n-Si is hard to determine.

2. In review 1 question 5 stability test, the author might want to analyze the catalyst performance at 120 h, where the degradation started, instead of 130 h. Moreover, the sample should be raised with water to get rid of the adsorbed NaOH before XPS analysis.

Reviewer #3 (Remarks to the Author):

Comments to “Atomically Dispersed Ir Catalysts on Si Photoanode for Efficient Photoelectrochemical Water Splitting”

The authors studied single atom Ir dispersed in NiO/Ni matrix as photoanodes in photoelectrochemical cell applications.

I have issue with the novelty of this work. Single atom catalysis has been some years and Ir/IrO₂ itself is known for being a good OER catalyst. It is not clear fundamentally what this work offers is significantly beyond current knowledge in order to publish on Nat. Commun.

More specifically, I have some questions to authors:

1. Ir can be substitution and interstitial sites; did the authors exam these options?
2. Ir is only stable in acidic condition; did the authors study pH dependence on this material?
3. The authors only computed thermodynamic reaction energies but did not study kinetics such as reaction barriers and rates; the connection with experimental measurements isn't clear.
4. Does the catalytical activity depend on Ir doping concentration?

Response to the Reviewers' Comments

We thank the reviewers for their constructive comments and suggestions that are helpful for us to improve the manuscript, entitled "*Atomically Dispersed Ir Catalysts on Si Photoanode for Efficient Photoelectrochemical Water Splitting*" (NCOMMS-22-17185A). We have fully revised the manuscript taking into account of the reviewer's comments. A point-by-point response to the reviewers' comments is given below.

Reviewer #1:

The authors have answered the questions asked the reviewers and improved the manuscript, this paper should be now publishable in Nat Commun.

Response: We are sincerely thankful for the time and effort that Reviewer 1 has spent in reviewing our manuscript.

Reviewer #2:

The author has thoroughly answered the reviewers' previous questions. However, there are two questions that need to be further clarified.

Response: We are grateful for the time and effort that Reviewer 2 has spent in reviewing our manuscript. Please see our point-by-point responses to the reviewer's comment.

1. In review 1 question 2, to confirm the role of ZrO_2 , it will be great to compare the saturation current differences of Ni/n-Si and Ni/ ZrO_2 /n-Si. Unfortunately, the saturation current of Ni/n-Si is hard to determine.

We sincerely thank the reviewer for the insightful comment. To clearly compare the saturation current difference of Ni/n-Si and Ni/ ZrO_2 /n-Si, the LSV curve of Ni/n-Si was newly obtained and the scale of the x-axis was increased. Due to the ZrO_2 layer, the saturation photocurrent density was increased by 2 mA cm^{-2} .

→ According to this comment, we have revised the manuscript and Fig. 3a.

In manuscript,

“As a result, the onset potential shifts toward the cathodic direction with the value of 1.14 V versus RHE to reach 1 mA cm^{-2} and the saturation photocurrent density was increased by 2 mA cm^{-2} .”

In Fig. 3a,

2. In review 1 question 5 stability test, the author might want to analyze the catalyst performance at 120 h, where the degradation started, instead of 130 h. Moreover, the sample should be raised with water to get rid of the adsorbed NaOH before XPS analysis.

We appreciate the reviewer for your helpful comment. We conducted XPS analysis before the degradation of the photoanode and the surface of catalyst was fully cleaned by water to get rid of adsorbed NaOH. As a result, it was confirmed that the chemical state of Ir element shows no particular difference compared with that of the sample before measurement.

→ According to this comment, we have revised the manuscript and Supplementary Fig. 18.

In manuscript,

It is attributed to a chemically robust NiO/Ni catalyst and its capability to stabilize and activate Ir single atoms through strong interactions. Before the degradation of performance, the chemical state of Ir single atoms remained unchanged (Supplementary Fig. 18).

In supplementary information,

Figure S18. a Ir 4f spectra of Ir SAs/NiO/Ni/ZrO₂/n-Si photoanode before performance degradation at 120 h.

Reviewer #3:

The authors studied single atom Ir dispersed in NiO/Ni matrix as photoanodes in photoelectrochemical cell applications. I have issue with the novelty of this work. Single atom catalysis has been some years and Ir/IrO₂ itself is known for being a good OER catalyst. It is not clear fundamentally what this work offers is significantly beyond current knowledge in order to publish on Nat. Commun. More specifically, I have some questions to authors:

Response: We are grateful for the time and effort that Reviewer 3 has spent in reviewing our manuscript. Please see our point-by-point responses to the reviewer's comment.

1. Ir can be substitution and interstitial sites; did the authors exam these options?

We appreciate the reviewer for your thoughtful comment. We considered whether the Ir atoms exist in either substitution or interstition and concluded that Ir single atoms substituted Ni sites. Single atoms can be anchored on the surface of metal oxide via strong metal-oxygen interactions at coordinatively unsaturated defects such as vacancies. The NiO intrinsically has lots of Ni vacancies suitable for the stabilization of single atoms. As a result, Ir single atoms substituted Ni sites by interacting with oxygen atoms. In Fig. 1d and e, the intensity profiles were demonstrated to identify the location of Ir atoms. It was clearly confirmed that the Ir atoms are located at exactly the same positions of Ni atoms with high periodicity, exhibiting that the original Ni sites were occupied by Ir atoms.

→ According to this comment, we have revised the manuscript.

In manuscript,

In Fig. 1d and e, the atomic line profiles were demonstrated and Ir atoms are distinguished by the brighter spots with high signal intensity. It was clearly confirmed that the Ir atoms were located at exactly the same positions of Ni atoms with high periodicity, exhibiting that the original Ni sites were substituted by Ir atoms.

2. Ir is only stable in acidic condition; did the authors study pH dependence on this material?

We thank the reviewer for the insightful suggestion. As the reviewer recommended, we obtained LSV curves of Ir SAs, NCs, and film deposited on NiO/Ni/ZrO₂/n-Si photoanodes in an acidic environment (0.5 M H₂SO₄) to analyze the pH dependence. Although Ir is stable in acidic condition as the reviewer mentioned, the Ni-based thin film which was used for anchoring Ir atoms is highly vulnerable to this environment, resulting in being rapidly etched. As a result, Ir SAs/NiO/Ni/ZrO₂/n-Si and Ir NCs/NiO/Ni/ZrO₂/n-Si photoanodes lost catalytic effects and can not serve as photo-harvesting devices. Meanwhile, Ir film is stable in acidic condition and protects the NiO/Ni from being etched, showing a substantial photoelectrochemical catalytic activity.

→ According to this comment, we have revised the manuscript and Supplementary Fig. 19.

In manuscript,

“To analyze the pH dependence of as-fabricated samples, the LSV curves of Ir SAs, NCs, and film deposited on NiO/Ni/ZrO₂/n-Si photoanodes are obtained in acidic condition (0.5 M H₂SO₄) (Supplementary Fig. 19). Since the Ni-based thin film is highly vulnerable to this environment, Ir SAs/NiO/Ni/ZrO₂/n-Si and Ir NCs/NiO/Ni/ZrO₂/n-Si photoanodes lost catalytic effects and can not serve as photo-harvesting devices. Meanwhile, Ir film is stable in acidic condition and protects the NiO/Ni from being etched, showing a substantial photoelectrochemical catalytic activity.”

“In acidic condition, a saturated calomel electrode (SCE), graphite rod, and 0.5 M H₂SO₄ were used.”

$$E(\text{RHE}) = E(\text{SCE}) + E^0(\text{SCE}) + 0.059 \times \text{pH} \quad (2)$$

“E(SCE) is the measured potential versus the SCE reference electrode through a potentiostat and E⁰(SCE) is 0.241 V at 25 °C.”

In supplementary information,

Figure S19. LSV curves of Ir SAs, NCs, and film deposited on NiO/Ni/ZrO₂/n-Si photoanodes in acidic condition.

3. The authors only computed thermodynamic reaction energies but did not study kinetics such as reaction barriers and rates; the connection with experimental measurements isn't clear.

We thank the reviewer for your valuable comments on the manuscript regarding the computed calculations. Previously, thermodynamic reaction energies of NiO (100), IrO₂ (110), and Ir SAs/NiO (100) were calculated to identify the potential determining step (PDS) and compare thermodynamic energy barrier. As a result, we revealed that Ir SAs/NiO (100) showed the lowest thermodynamic energy barrier at the step of conversion from O* to OOH*.

As the reviewer mentioned, thermodynamic model may alone be insufficient to observe the kinetic reaction. In accordance with this, we additionally examined the kinetics of Ir SAs/NiO catalyst to predict the barrier heights of rate-determining step via climbing image nudged elastic band (CI-NEB) calculations. In a CI-NEB calculation, the reaction path can be investigated by spring-connected beads correlating with adjacent molecular configurations.

As the rate-determining step is the formation of OOH*, CI-NEB calculation was conducted from O* + OH (initial state) to OOH* (final state). Calculated with the most favorable incident OH angle, the molecular configurations at each state were demonstrated and corresponding energy values were obtained. In the transition state where there is a repulsion between O* and OH, a small kinetic barrier of about 0.05 eV is observed.

Since the reaction may occasionally occur at other incident OH angles, calculations were carried out for these cases. They show slightly increased kinetic energy barriers because OH requires additional energy as it rotates for a favorable incident angle. Considering other studies on CI-NEB calculations of active OER catalysts^{1, 2}, these results are reasonable to account for the kinetics of Ir SAs/NiO catalyst.

→ According to this comment, we have revised the manuscript, Supplementary Fig. 24, and 25.

In manuscript,

“In addition, the reaction kinetics of Ir SAs/NiO (100) was examined to predict the kinetic barrier heights of the rate-determining step via climbing image nudged elastic band (CI-NEB) calculations in Supplementary Fig. 24. Calculated with the most favorable incident OH angle, a small kinetic barrier of about 0.05 eV is observed in the transition state where there is a repulsion between O* and OH. In Supplementary Fig. 25, the reaction energy profiles and energy barriers at other incident OH angles are provided. They show slightly increased kinetic energy barriers because OH requires additional energy as it rotates for a favorable incident angle.”

“To predict kinetic barriers of the transition state at the potential determining step, the climbing image nudged elastic band method was adopted⁵⁸. Damped molecular dynamics and quick-min force-based optimizer were used and the self-consistent calculations of single-electron wavefunction were terminated when the iterative convergence of energy and force fulfilled 10^{-4} eV and 0.1 eV/\AA , respectively.”

“58. Henkelman, G., Uberuaga, B. P. & Jónsson, H. A climbing image nudged elastic band method for finding saddle points and minimum energy paths. *J. Chem. Phys.* **113**, 9901-9904 (2000).”

In supplementary information,

Figure S24. Reaction energy profile of Ir SAs/NiO (100) for *OOH formation calculated with the most favorable incident OH angle.

Figure S25. Reaction energy profiles of Ir SAs/NiO (100) for *OOH formation calculated with different incident OH angles.

4. Does the catalytic activity depend on Ir doping concentration?

We thank the reviewer for the insightful suggestion. It is thought that the catalytic activity will be higher as Ir doping concentration is increased while maintaining the single atom form. As the active center of the as-synthesized catalyst is the isolated Ir single atom enabling the effective photogenerated charge transport and lowering the energy barrier in the potential-determining step, the reaction will be boosted on the enlarged active sites.

We newly synthesized the Ir catalyst via 2 cycle-ALD process and the photoelectrochemical catalytic activity of it was compared to that of Ir SAs (1 cyc.), which could confirm how the PEC performance changes depending on Ir concentration. However, the catalytic activity of Ir (2cyc.) was not higher than that of Ir SAs (1cyc.). This is because Ir deposited in the second cycle adheres to the existing single atoms rather than forming new single atoms. As a result, even if the Ir concentration increases, the catalytic activity does not increase unless new single atoms are formed.

→ According to this comment, we have revised the manuscript and Supplementary Fig. 13.

In manuscript,

“In Figure 3b and Supplementary Fig. 13, the photoanode with Ir SAs shows the lowest onset potential and the highest photocurrent density over the whole potential range among all samples, implying that Ir SAs exhibit the highest catalytic activity and photon harvesting.”

In supplementary information,

Figure S13. LSV curves of the photoanodes with Ir SAs and Ir catalyst synthesized via 2-cycle ALD process.

The catalytic activity of Ir (2cyc.) was not higher than that of Ir SAs (1cyc.). This is because Ir deposited in the second cycle adheres to the existing single atoms rather than forming new single atoms. As a result, even if the Ir concentration increases, the catalytic activity does not increase unless new single atoms are formed.

References

- 1) Partanen, L., Murdachaew, G. & Laasonen, K. *J. Phys. Chem. C* **122**, 12892-12899 (2018).
- 2) Dickens, C. F., Kirk, C. & Norskøv, J. K. *J. Phys. Chem. C* **123**, 18960-18977 (2019)